# Collective buoyancy-driven dynamics in swarming enzymatic nanomotors

Shuqin Chen [1,2], Xander Peetroons[1], Anna C. Bakenecker [1], Florencia Lezcano [1], Igor S. Aranson [3,4,5] ✉ & Samuel Sánchez [1,6] ✉

Enzymatic nanomotors harvest kinetic energy through the catalysis of chemical fuels. When a drop containing nanomotors is placed in a fuel-rich environment, they assemble into ordered groups and exhibit intriguing collective behaviour akin to the bioconvection of aerobic microorganismal suspensions. This collective behaviour presents numerous advantages compared to individual nanomotors, including expanded coverage and prolonged propulsion duration. However, the physical mechanisms underlying the collective motion have yet to be fully elucidated. Our study investigates the formation of enzymatic swarms using experimental analysis and computational modelling. We show that the directional movement of enzymatic nanomotor swarms is due to their solutal buoyancy. We investigate various factors that impact the movement of nanomotor swarms, such as particle concentration, fuel concentration, fuel viscosity, and vertical confinement. We examine the effects of these factors on swarm self-organization to gain a deeper understanding. In addition, the urease catalysis reaction produces ammonia and carbon dioxide, accelerating the directional movement of active swarms in urea compared with passive ones in the same conditions. The numerical analysis agrees with the experimental findings. Our findings are crucial for the potential biomedical applications of enzymatic nanomotor swarms, ranging from enhanced diffusion in bio-fluids and targeted delivery to cancer therapy.

Collective behaviour is widespread in nature. While individual units of a group obey simple rules, they present complex and intriguing collective behaviour when assembling into highly ordered structures[1]. Living organisms use distributed or swarm intelligence to accomplish sophisticated tasks to survive. Examples range from collective cell migration[2], honeybees adapting to repeated shaking to maintain mechanical stability of the swarm[3], to emperor penguins packing in a huddle in a highly coordinated manner to survive cold winter[4]. Multiple synthetic swarming systems have been developed with inspiration from nature including: (1) applying one or multiple external forces,

such as magnetic fields[5–8], light[9,10], ultrasound[11,12], electric fields[13,14], (2) utilizing chemicals as signals[15–17], (3) combining biological microswimmers, such as sperm cells and algae, into artificial moieties as a hybrid integration[18–20], (4) exploiting DNA base-pair interactions[21,22]. These well-designed swarms show many advantages compared to single-unit functionalities, like enhanced coverage and fluid mixing, intelligent multitasking, collective chemotaxis and perception, and environmental adaptation.

Micro/nanomotors (MNMs) are synthetic active devices achieving self-propulsion through converting various types of energy into

[1]Institute for Bioengineering of Catalonia (IBEC), The Barcelona Institute for Science and Technology (BIST), Baldiri i Reixac 10-12, Barcelona 08028, Spain. [2]Faculty of Physics, University of Barcelona, Martí i Franquès 1, Barcelona, Spain. [3]Department of Biomedical Engineering, The Pennsylvania State University, University Park, PA 16802, USA. [4]Department of Chemistry, The Pennsylvania State University, University Park, PA 16802, USA. [5]Department of Mathematics, The Pennsylvania State University, University Park, PA 16802, USA. [6]Catalan Institute for Research and Advanced Studies (ICREA), Passeig Lluís Companys 23, Barcelona, Spain. ✉e-mail: isa12@psu.edu; ssanchez@ibecbarcelona.eu

mechanical motion[23,24]. Earlier works on enzyme-powered MNMs have demonstrated the motion of single particles[25–27] and small clusters[28], as well as proof-of-concept studies in drug delivery[29–33] and sensing[34,35]. Nonetheless, recent reports have shifted focus to the collective motion of these particles. Recently, Hortelao et al.[36] reported the emergent swarming behaviour of enzymatic nanomotors. The urease-powered nanomotors show collective migration in urea, demonstrating the ability to swim across complex paths compared to the inactive nanomotors. Furthermore, the active collective dynamics, combined with advanced imaging technologies, position them as promising tools in the field of biomedicine. For example, swarms of radio-labelled nanobots have shown an eightfold increase in tumour penetration and approximately a 90% reduction in tumour size during radionuclide therapy[37]. Swarms of catalase-powered nanobots overcome and disrupt mucus layer, resulting in a 60-fold increase in mucus barrier penetration, through in vitro and ex vivo validation[38]. Hyaluronidase and urease nanomotor swarms work synergistically for enhanced diffusion in viscous media, such as synovial fluid, paving the way for treating joint injuries[39]. Similarly, collagenase-powered MNMs[40,41] and urease-powered iron oxide nanomotor swarms[42] were exploited to disrupt collagen fibres, serving as a model of the extracellular medium. This disruption facilitates cell spheroids penetration and enhances the delivery efficiency of a second swarm of nanomotors by 10-fold.

Although enzyme-powered MNMs have primarily demonstrated their potential in biomedical applications, the mechanisms underlying the emergent collective behaviour remain to be clarified. Inspired by nature, the intriguing collective phenomenon bears a resemblance to bioconvection. Bioconvection is a self-organized and self-sustained vortex motion that arises naturally in suspensions of microorganisms[43]. It visually resembles the Rayleigh–Bénard convection in fluid heated from below[44]. The bioconvection emerges due to the unstable density gradients resulting from the accumulation of buoyant microorganisms[45]. Each microorganism plays a pivotal role in driving accumulation and fluid flow. Certain gravitactic algae or aerotactic bacteria exhibit upward swimming. In the presence of an upper surface, they form a thin boundary layer of microorganism-rich heavier fluid, which becomes unstable, leading to the formation of falling plumes[46].

In synthetic MNMs, buoyancy-driven convection has been employed for directional motility and cargo delivery. One approach utilizes incident light to generate convective flow through the photothermal effect. This convective flow can drive $TiO_2$ micromotors to aggregate and form clusters[47,48], or enable magnetic colloidal collectives to drift using fluidic currents[49]. Another approach involves enzymes fixed on a surface, which catalyze fuels, inducing density variations between the reactants and products of chemical reactions. For instance, urease-attached macroscale sheets exhibit clockwise or

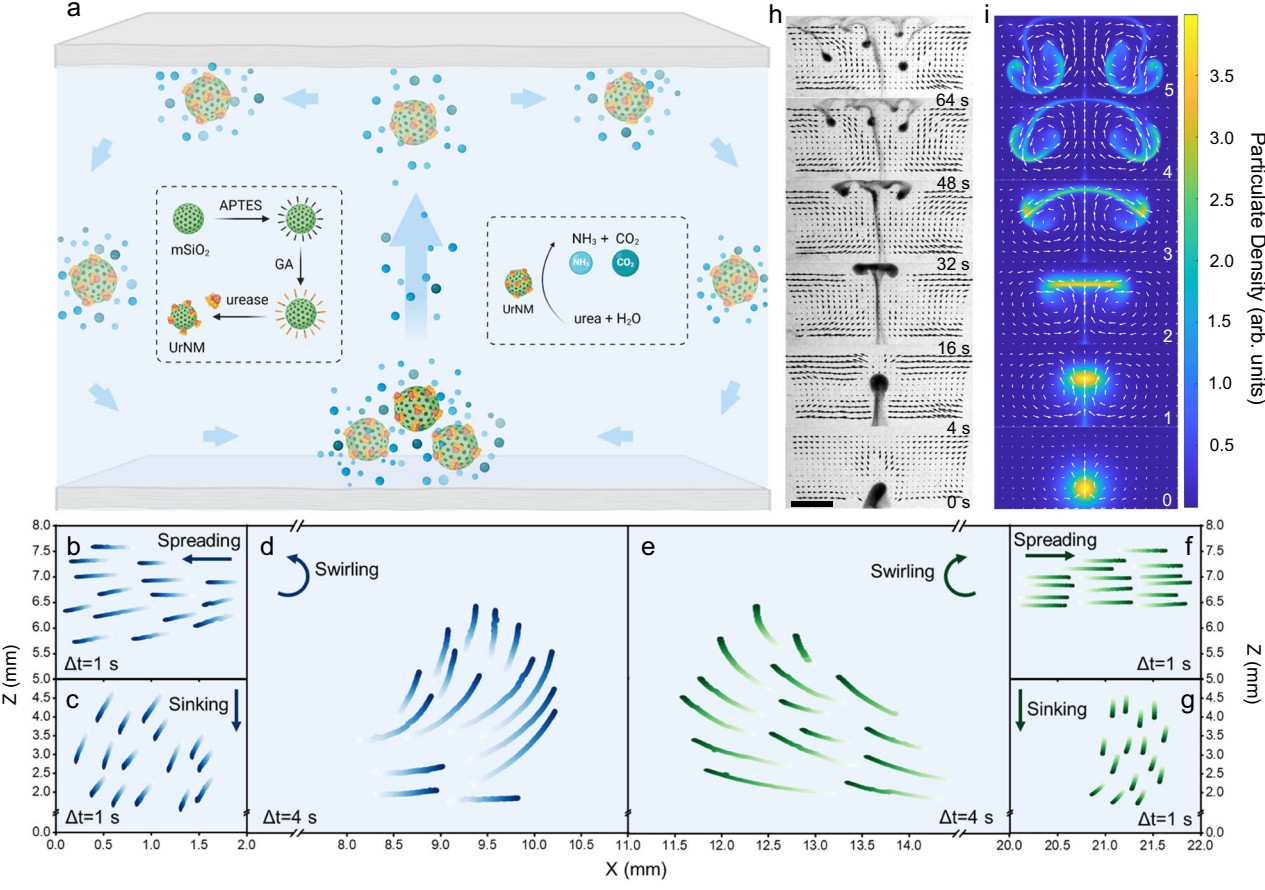

**Fig. 1 | Collective behaviour of enzymatic nanomotors viewed from the side. a** Schematics illustrating the preparation of enzymatic nanomotors and the mechanism of solutal buoyancy resulting in collective behaviour. Created in BioRender. Sánchez, S. (2023) BioRender.com/b39h124. **b**–**g** Trajectory tracking of the UrNMs in collective movement. **b** and **f** depict one-second trajectories during a spreading stage. **c** and **g** show one-second trajectories during a sinking stage. **d** and **e** display four-second trajectories during a swirling stage in urea. The blue and green colour-coded trajectories indicate counterclockwise and clockwise directions of UrNMs on the left and right sides of the chamber, respectively. $N = 15$. **h** A time-lapse sequence of images that show the directional and collective movement of enzymatic nanomotors in fuel. The fluid flow is analyzed by adding tracer particles and is shown in black arrows. Scale bar: 4 mm. **i** A time-lapse sequence of snapshots of computational results according to the assumed mechanism. The colour bar indicates the nanomotor concentration, and the white arrows display the fluid velocity.

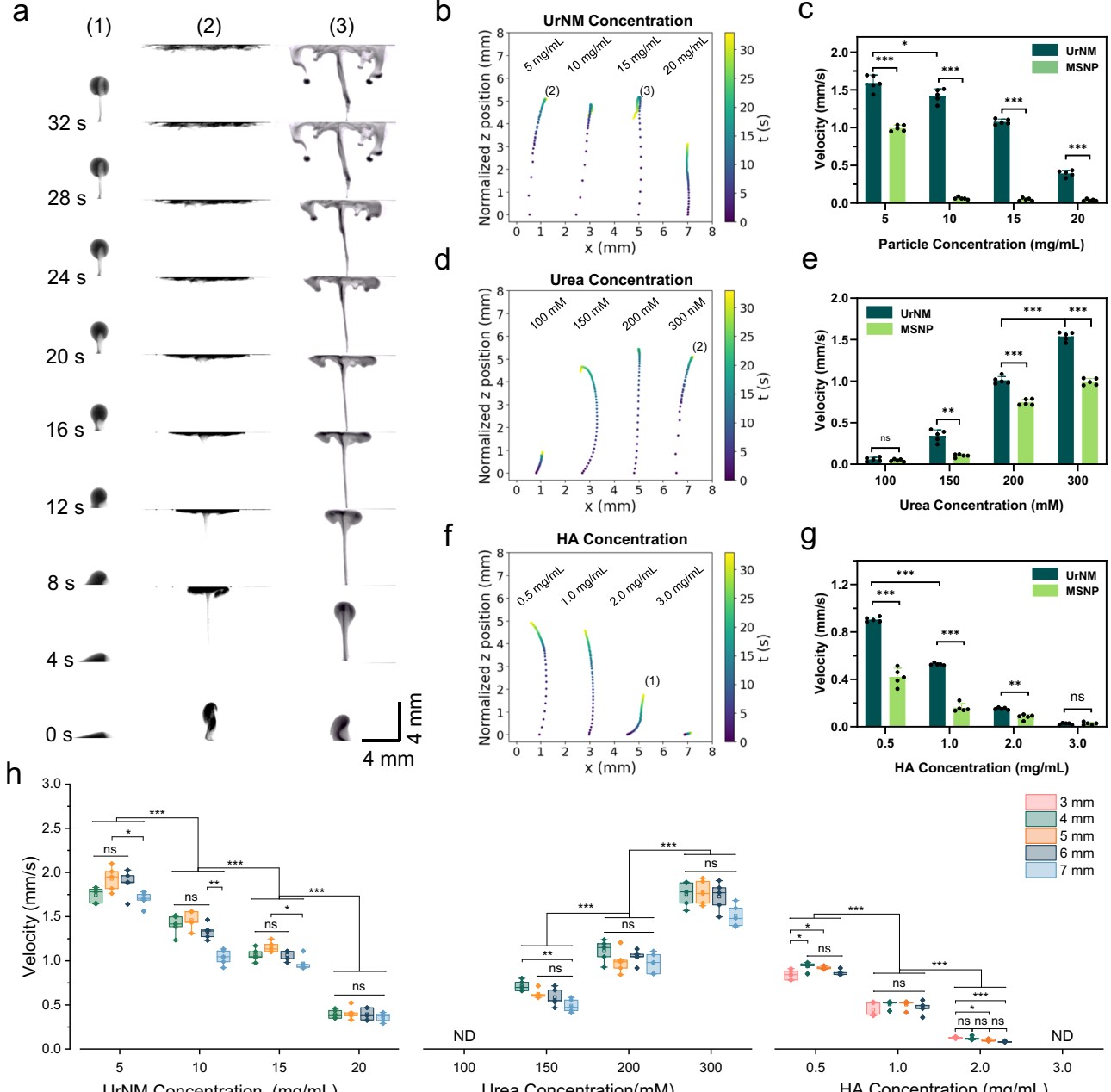

**Fig. 2 | Control factors that affect collective behaviour. a** A time-lapse sequence of images capturing the movement of enzymatic nanomotors in fuel, showing three different stages of collective behaviour, i.e. (1) ascending, (2) ascending and spreading, (3) ascending, spreading, and sinking. Scale bar: 4 mm × 4 mm. The centre of mass tracking of UrNMs swarms in z-axis under various conditions: **b** UrNM concentration, **d** urea concentration, and **f** HA concentration. (1)-(3) are chosen and displayed in **a**. Velocity analysis of active UrNMs swarms and passive MSNPs particulates shown in **c**, **e** and **g** correspond to **b**, **d** and **f**, respectively. The data are shown as mean velocity ± standard deviation (SD) of five independent experiments ($N = 5$). **h** Velocity analysis in z-axis at different heights during particulates' upward movement in varied UrNM concentration, urea concentration, and fuel with diverse HA concentration. The box represents the interquartile range, the whiskers represent the minimum and maximum values, and the central line represents the median. Significant difference is analyzed by student's t-test: ***$P < 0.001$; **$P < 0.01$; *$P < 0.05$; ns = not significant ($P > 0.05$). $N = 5$. ND means velocity is lower than the detectable value.

counterclockwise rotation via a solutal buoyancy mechanism arising from enzyme pumps[50]. Additionally, urease attached to a lipid membrane induces fluid flow and the directional transport of tracer particles[51]. These studies underscore the significant implications of convective flow on the collective movement of micro/nanoparticles.

Here, the buoyancy-driven convective dynamics of collective enzymatic nanomotors arises from the density difference between the product-rich particulate and the denser environment. We attach enzymes onto the surface of nanoparticles, which move dynamically with the fluid flow. We describe the emergent collective behaviour of enzymatic nanomotor in three dimensions (3D) to unveil the underlying mechanisms. We model a swarm as light particulates immersed in a denser fluid environment. The particulate swarm moves upward, creating a convective flow in a closed fuel-filled space. Control factors for directional and collective mobility of enzymatic nanomotors, such as particle and fuel concentration and media viscosity, are investigated. In 3D space, convective flows develop complex patterns such as vortices. To further validate our findings, we vertically confine the system to two dimensions, where convective flow is constrained to a plane, inhibiting the possible flow configurations. The convective

dynamics resemble bioconvection and provide insights into the mechanisms of collective behaviour observed in enzymatic nanomotors.

## Results and Discussion

In our study, we view the collective behaviour of enzymatic nanomotors from the side or from the top. The enzymatic nanomotors are based on mesoporous silica nanoparticles with urease attached (UrNMs). Detailed characterization can be found in the Supplementary Materials (Fig. S1) and the Methods section. The molecularly unbalanced distribution of enzymes generates net motion for single nanomotors[52,53] in urea, Fig. S2. From the side, upon introducing a drop of particulate in a fuel-filled chamber, the drop shows upward motion against gravity, generating a convective flow within the closed space, Fig. 1a. During an ascending stage, there are two counter-rotating vortices within a droplet. A characteristic hydrodynamic flow pattern is displayed in Fig. S3. As the particulate reaches the upper boundary, it spreads to balance the mean upward force, forming a layer of unstable particle-rich fluid. The layer then sinks in the form of falling plumes. Trajectory tracking of UrNMs in a $22 \times 8 \times 1.6$ mm (length × height × width) chamber shows the spreading, sinking, and swirling stages of the convective dynamics, Fig. 1b–g. These trajectories on the left and right are not perfectly symmetric due to experimental limitations. The upward movement of a nanomotor swarm is due to buoyancy arising from the density difference between the reaction product-rich particulate and the media with fuel. We state that individual nanomotors perform urease catalysis reaction and generate ammonia and carbon dioxide, making the particulate lighter. Since the temperature change during chemical reactions is not obvious (Fig. S4), we rule out heat effect on the upward movement. In addition, the analysis of enzymatic activity and the upward velocity of UrNMs in urea at physiological temperature (37 °C) shows no significant difference compared to room temperature (25 °C), Fig. S5a–c. We performed computational modelling based on two-fluid hydrodynamics and compared the computational results to the experiments; a good qualitative agreement is obtained, see Fig. 1h, i.

To verify the universality of our mechanisms, we synthesized catalase-powered nanomotors (CatNMs) and observed the convective dynamics of these enzymatic nanomotors in hydrogen peroxide $(H_2O_2)$ (Fig. S6 and video S1). Notably, the instant chemical reaction of CatNMs in $H_2O_2$ results in a burst of oxygen bubbles, which drives the CatNMs to move upward against gravity within one second. We expect that this buoyancy-driven mechanism will also apply to other asymmetric motors, such as Janus motors. Previous studies in our group have verified the difference between the Janus structure and the patchy-like structure. The motion of a single out-of-equilibrium particle arises from the asymmetric distribution of ions, which generates an ionic gradient[54]. A theoretical study has shown that Janus particles exhibit higher velocities compared to patchy-like motors[55]. Additionally, it has been reported that micron-sized hollow urease motors present a 3D motion at a single particle level[56], thus it is expected that large populations of these particles will also show collective motion. However, if the fabrication of Janus structures involves heavier materials like platinum or gold, sedimentation may neglect buoyancy[57–59], impeding the upward collective movement. That suggests that the buoyancy-driven mechanism could be universal for various types of motors across different length scales, provided that the gravitational effects would not suppress the buoyancy-driven motion.

### Controlling collective behaviour of UrNMs

Collective behaviour can be viewed from the side. We studied the influence of three main control factors, UrNM concentration, urea concentration, and viscosity mediated by hyaluronic acid (HA) concentration, on the collective behaviour. As illustrated in Fig. 2a, there are three stages of the collective behaviour of enzymatic nanomotors,

i.e., ascending (1, 2 and 3), spreading (2 and 3), and sinking (3). When a swarm of nanomotors seeded in the bottom of the chamber that is filled with fuel, they show directional mobility against gravity. Under different conditions, they display various forms of collective behaviour and velocity. Figure 2b shows the $z$-component of the particulate centre of mass as a function of time within 32 s. The velocity difference can be deduced from Fig. S7 showing that for higher UrNM concentrations, particulates reach a lower $z$ position in 5 s. As the UrNM concentration increases to 20 mg/mL, the majority of the nanomotor swarms cannot get to the upper boundary due to gravity (Fig. 2b, c and video S2). The velocity field was analyzed by front-tracking the particulate based on custom Python code. As expected, compared with passive nanoparticles (MSNPs), active nanomotors show enhanced upward speeds, Fig. S8 and video S3. We assume that enzymatic catalysis of urea produces microbubbles[60] and the product, ammonia, makes this particulate less dense. Although the product quantity may be larger with higher UrNM concentration, the density of particulate increases as well when we increase the concentration of nanoparticles. We state that there should be a competition between the two opposite conditions, after which the effect of increased particulate density takes the lead, and the upward particulate velocity decreases with the increased UrNM concentration.

Buoyancy, the main driving force, is strongly influenced by fuel concentration. Figure 2d, e show that the upward speeds increase with the fuel concentration. One can clearly observe the upward motion of particulates at concentrations of 150 mM urea and above. However, in the presence of 100 mM urea concentration, particulate almost stays at the seeding point, and there is no difference between the upward motion of active swarms and passive particulates. We argue that this is because in low urea concentration, density difference resulting in a buoyancy force is not sufficient to lift the particulate. To distinguish the behaviours between active nanomotors and passive nanoparticles, we introduce phenol red, a pH indicator, into the urea solution. The chemical reactions that occur during the collective movement of active nanomotors lead to a pH change in the surrounding solution, resulting in a colour shift from light yellow to pink, video S4 and Fig. S9. This colour change indicates the location of the active UrNMs. For comparison, passive nanoparticles were tested in the same condition (10 mg/mL, 200 mM urea). They expand along the bottom plane at time 0, and the pH of their surrounding solution remains unchanged.

We added hyaluronic acid into the fuel to change the media viscosity observing that the upward speeds of particulate decreases with the increase in concentration of hyaluronic acid, Fig. 2f, g. As it was shown above, active swarms show enhanced speed compared to passive particulates in viscous media. When the concentration of hyaluronic acid increases to 3 mg/mL, both active swarms and passive particulates remain at the seeding point because higher viscosities inhibit fluid convection. We conducted particulate velocity analysis at elevated heights in the middle of the chamber. In Fig. 2h, active particles move slightly faster in the middle of their paths and decrease their speeds when approaching the upper boundary in different groups, while passive particles keep decreasing their speeds (Fig. S10). For instance, a particulate of 5 mg/mL UrNMs moves upward at $1.74 \pm 0.09$ mm/s at 4 mm height, $1.93 \pm 0.14$ mm/s at 5 mm height, and $1.70 \pm 0.08$ mm/s at 7 mm height, while a particulate of the same concentration of passive nanoparticles moves at $0.96 \pm 0.03$ mm/s at 4 mm height, $0.68 \pm 0.02$ mm/s at 5 mm height, and $0.47 \pm 0.02$ mm/s at 7 mm height. The acceleration process of active particulate could be due to the density changes caused by chemical reaction products. Additionally, during the spreading stage, active nanomotors form a thin boundary layer of particle-rich fluid that continues to spread until it meets the side boundary. In contrast, passive particles form a less stable boundary layer, leading to the formation of larger falling plumes earlier, causing them to sink before reaching the side boundary (see

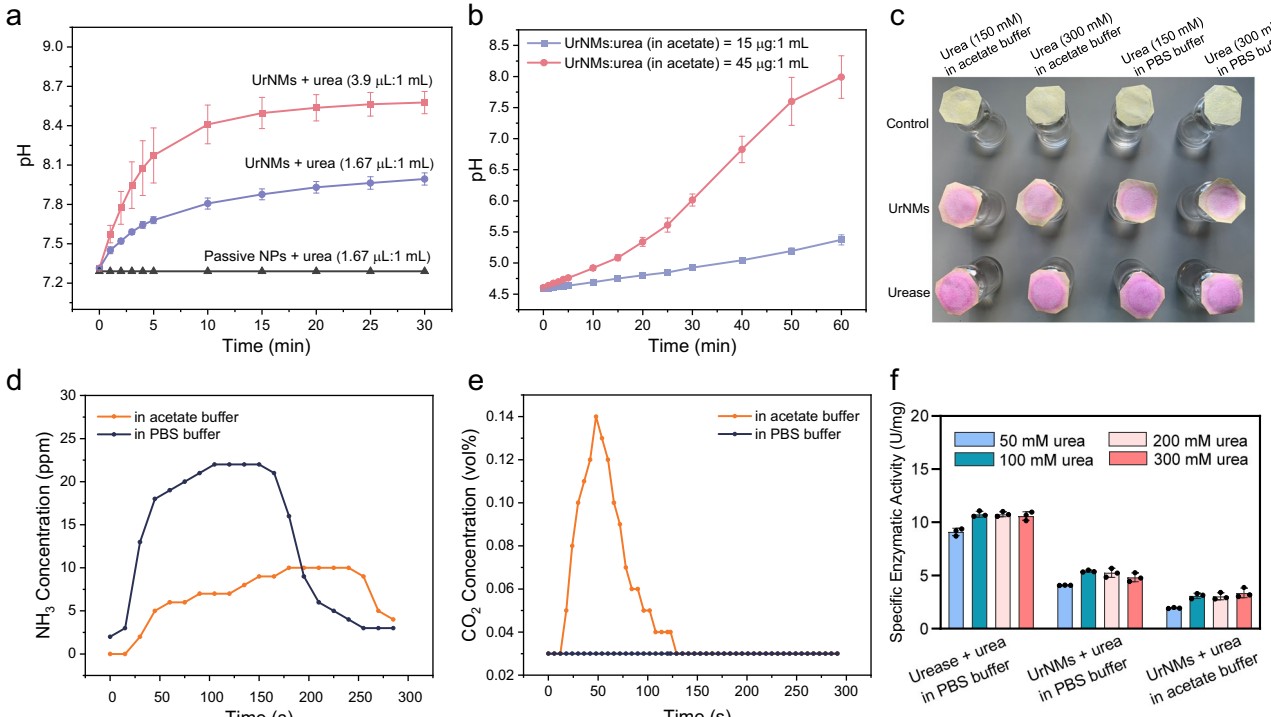

**Fig. 3 | Verification of UrNMs catalysis reaction products.** Real-time pH changes with time when adding varied amount of UrNMs into **a** urea dissolved in PBS buffer and **b** urea dissolved in acetate buffer. The data are shown as mean ± SD of three independent experiments ($N = 3$). **c** UrNMs or urease were added in urea dissolved in PBS buffer or acetate buffer (300 mM or 150 mM). Nothing was added in the control group. The colour change of phenol red indicates the production of ammonia. **d** Real-time monitoring of $NH_3$. **e** Real-time monitoring of the generated $CO_2$. **f** The specific enzymatic activity of urease and UrNMs in different concentrations of urea dissolved in PBS or acetate buffer. $N = 3$.

videos S2 and S3). We assume this occurs because the products of chemical reactions make the active particles less dense, and the faster upward movement of active nanomotors creates a more dynamic environment with increased fluid flow, making it less likely to form large falling plumes.

**Products of UrNMs catalysis reaction accelerate the directional movement**

Urease catalyzes the decomposition of urea into ammonia ($NH_3$) and carbon dioxide ($CO_2$). On the one hand, $NH_3$ is highly soluble in water due to the formation of hydrogen bonds with water molecules. This interaction results in a smaller density of the solution[61,62]. On the other hand, the released $NH_3$ dissolves in water, resulting in an alkaline solution (Fig. 3a) and promoting $CO_2$ to dissolve. Under proper fuel concentration, the formation of $NH_3$ and $CO_2$ microbubbles can be observed[60]. However, in acidic buffers, such as acetate buffer (pH = 4.6, Fig. 3b), $CO_2$ may exist because the abundant hydrogen ions inhibit the dissolution of $CO_2$ and the ionization of carbonic acid. The main reaction rate constants for $CO_2$ and $NH_3$ in phosphate buffer saline (PBS) buffer and acetate buffer are presented in table S1. In PBS buffer, the rate constant for $CO_2$ dissolution[63] in basic solutions ($k_1 = 1.21 \times 10^4$ $M^{-1}s^{-1}$) is much higher than the reverse rate constant ($k_{-1} = 4.0 \times 10^{-4}$ $s^{-1}$). In acetate buffer, ammonia dissolves in acidic solutions. The rate constant for $CO_2$ dissolution in water ($k_4 = 0.037$ $s^{-1}$) is much smaller than the reverse constant of $HCO_3^-$ combining with $H^+$ ($k_{-4} = 1.24 \times 10^5$ $M^{-1}$ $s^{-1}$).

We conducted experiments to verify the existence of $NH_3$ and $CO_2$. In Fig. 3c, cover papers were pre-dipped in phenol red solutions. Upon adding urease or UrNMs into urea solution, $NH_3$ is produced and volatilizes until it dissolves in the cover paper that contains phenol red, the colour change of which from light yellow to pink indicates the presence of $NH_3$. The production of $CO_2$ can be observed in acetate

buffer, which maintains an acidic environment during the urease catalysis reaction, Fig. 3b. $CO_2$ bubbles produced by UrNMs reacting with urea dissolving in acetate buffer can be observed on the wall of a cuvette (video S5). We filled cuvettes with 300 mM urea solutions that were dissolved either in PBS buffer or in acetate buffer. Then UrNMs or urease solutions were added to the cuvettes, respectively. Video S5 shows clearly the convective flow from the turbidity while for smaller urease molecules, the solutions remain transparent. In addition, the produced $NH_3$ and $CO_2$ in acetate buffer can be directly detected by a gas sensor, an optoelectronic analysis equipment that is able to accurately detect low-concentration gases at the ppm level, as shown in Fig. 3d, e.

The enzymatic activity of UrNMs in urea solutions in both PBS buffer (Fig. 3f, S11) and acetate buffer (Fig. S12) was examined. In PBS buffer, the specific enzymatic activity of UrNMs increases from $4.08 \pm 0.02$ U/mg in 50 mM urea solutions to $4.82 \pm 0.41$ U/mg in 300 mM urea solutions. In acetate buffer, the specific enzymatic activity of UrNMs is slightly weaker, with $1.94 \pm 0.05$ U/mg in 50 mM urea solutions and $3.36 \pm 0.45$ U/mg in 300 mM urea solutions. This is because the known optimum pH for urease catalytic activity is around 7 - 8[64]. We also examined the enzymatic activity of UrNMs in urine. Figure S13 shows that the enzymatic activity of urease decreases from $5.25 \pm 0.06$ U/mg to $3.56 \pm 0.15$ U/mg in simulated urine[65], and to $3.57 \pm 0.20$ U/mg in real urine. Although there is a significant decrease in enzymatic activity in urine for 30 min, the activity of UrNMs is still relatively high, verifying the potential of UrNMs for in vivo applications. The above results indicate that the urease catalysis reaction produces dissolved $NH_3$ and $CO_2$ in PBS buffer and dissolved $NH_3$ and $CO_2$ gas in acetate buffer. Therefore, upon quantifying the upward velocity of the UrNMs particulate, the results indicate a faster upward movement, from $1.01 \pm 0.04$ mm/s in PBS buffer to $1.14 \pm 0.04$ mm/s in acetate buffer, as shown in Fig. S14. These findings validate our

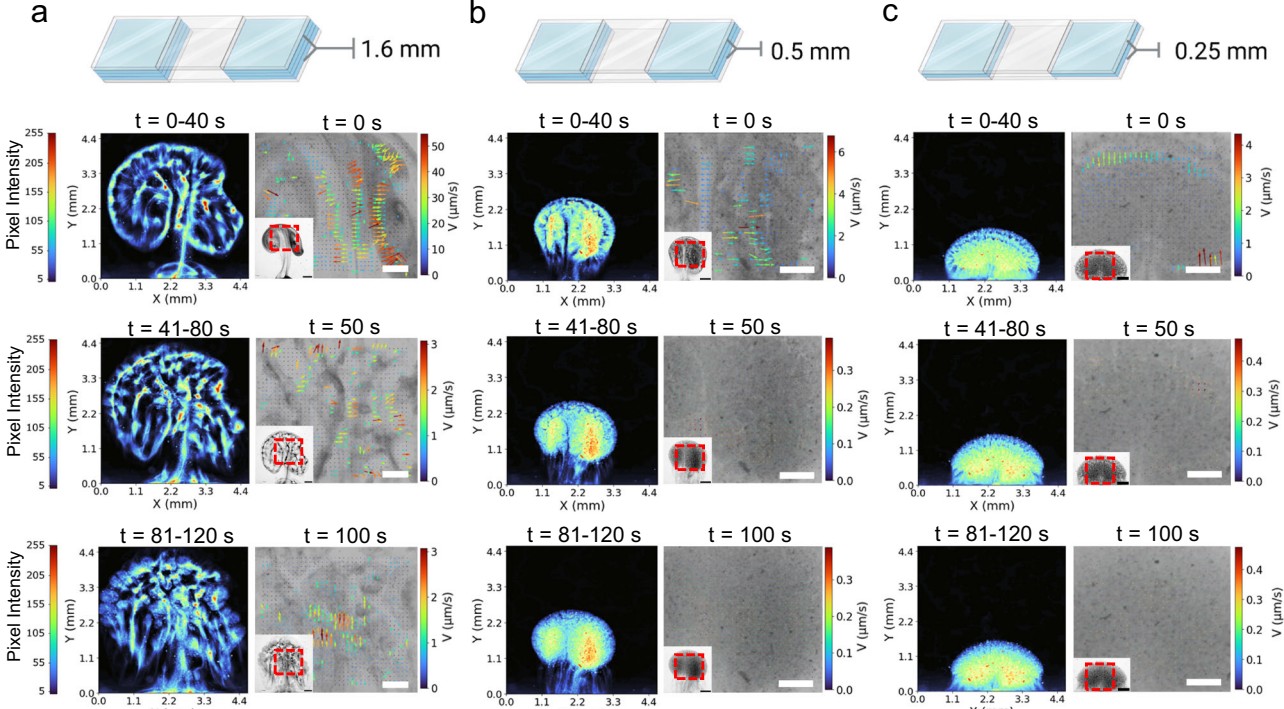

**Fig. 4 | Collective behaviour shaped by vertical confinement.** Intensity maps and particle image velocimetry (PIV) of UrNMs swarms in 300 mM urea solutions in microfluidic chips with varied heights of **a** 1.6 mm, **b** 0.5 mm, and **c** 0.25 mm. The average pixel intensity was calculated over 40 s periods from video recordings (left panels). A zoomed-in view in the right panels shows corresponding PIV measurements. Scale bars in the small panel: 1 mm, in the enlarged panel: 0.5 mm. The schematics were created in BioRender. Sánchez, S. (2023) BioRender.com/i44j104.

assumption that the chemical products result in a higher density difference between the particulate and the media with fuel, leading to an accelerated movement.

## Vertical confinement shapes collective behaviour

Since buoyancy is the primary force that drives the self-organization of active particulates, we studied the influence of vertical confinement on their collective behaviour. As shown in Fig. 4, microfluidic chips with three different heights (1.6 mm, 0.5 mm, and 0.25 mm) were designed and filled with urea in the vertically confined chamber. Then active UrNMs were introduced and entered the chamber from the side by capillary force. In Fig. 4a and video S6, these active UrNMs swarms exhibit collective movement in the chamber of 1.6 mm height. The density maps, observed from the top, show that the swarms aggregate, coarsen, and change their patterns over time. Particle image velocimetry (PIV) also confirms that the fluid flow is initially faster when the nanomotors are injected into the chamber. Fig. S15–S17 show the PIV results at 25 s time intervals in confinement with different heights. After 50 s, nanomotors keep moving and collective behaviour is still transient. After 100 s, the fluid flow keeps a relatively high speed, 1.5 μm/s on average. However, the fluid flow direction remains the same according to the arrows. As a comparison, without fuel UrNMs sink to the bottom in a confined chamber and expand along the bottom plane, Fig. S18. The convective flow is also weaker than that caused by UrNMs with fuel, Fig. S19–21. When the vertical confinement is changed to 0.5 mm, the movement of active UrNMs becomes localized. In Fig. 4b, the density map shows that the pattern of UrNMs only slightly changes over time. The PIV reveals that fluid flow velocity decreases compared to larger height values. After 50 s, the swarms barely move. When the height is further reduced to 0.25 mm, the swarms' movement is hindered, as displayed by the unchanged shape of swarms over time and the decreased velocity of fluid flow in PIV, Fig. 4c. Active UrNMs in PBS solutions also show decreased velocity when the chamber height

decreases (Fig. S21). However, compared with the active UrNMs in fuel, there are no significant differences. We also analyzed the swarm dynamics by pixel intensity distribution. A time-lapse sequence of snapshots at 12 s time intervals from video recordings is selected. As shown in Fig. S22, in a 1.6 mm-high chamber, the pixel intensity of active UrNMs in fuel is broadly distributed in the region of interest (ROI) in the initial 60 seconds, and gradually changes to narrowly distributed in 2 min. However, for the 0.5 mm-high chamber and the 0.25 mm-high chamber, pixel intensities are monodispersed in the ROI within the time durations. As a comparison, the pixel intensities of active UrNMs in PBS solutions are highly monodispersed in the three different chambers, Fig. S23. These results indicate that the vertical confinement controls the swarms by affecting fluid convective flows and provide insight into the buoyancy-driven collective behaviour of nanomotors.

## Computational modelling shows similarity with experiments

Our starting point is two-fluid hydrodynamics[66]. One fluid is a solvent with the kinematic viscosity $\eta$, flow velocity $\mathbf{v}$, solvent pressure $p$, and solvent density $\rho_0$. Second fluid is the particulate with the volume density $\rho$, coarse-grained particulate velocity $\mathbf{u}$, and pressure $P = q\rho$, and the factor $q$ depends on the temperature (as for gases). We describe the dynamics by the simplified Navier-Stokes Eq. (1), coupled to the reaction-advection equation for the concentration of chemical fuel $c$, Eq. (2), and a mass transport equation for the particulate density, Eq. (3):

$$\rho_0\left(\partial_t\mathbf{v} + \mathbf{v}\nabla\mathbf{v}\right) = \eta\nabla^2\mathbf{v} - \nabla p - \mathbf{z}_0\rho(g\alpha - \epsilon c) \tag{1}$$

$$\partial_t c + \nabla\cdot(\mathbf{v}c) = D_c\nabla^2 c - \gamma\rho c \tag{2}$$

$$\partial_t\rho + \nabla\cdot(\mathbf{v}\rho) = \left(q\nabla^2\rho + \alpha g\partial_z\rho\right)/\kappa_1 \tag{3}$$

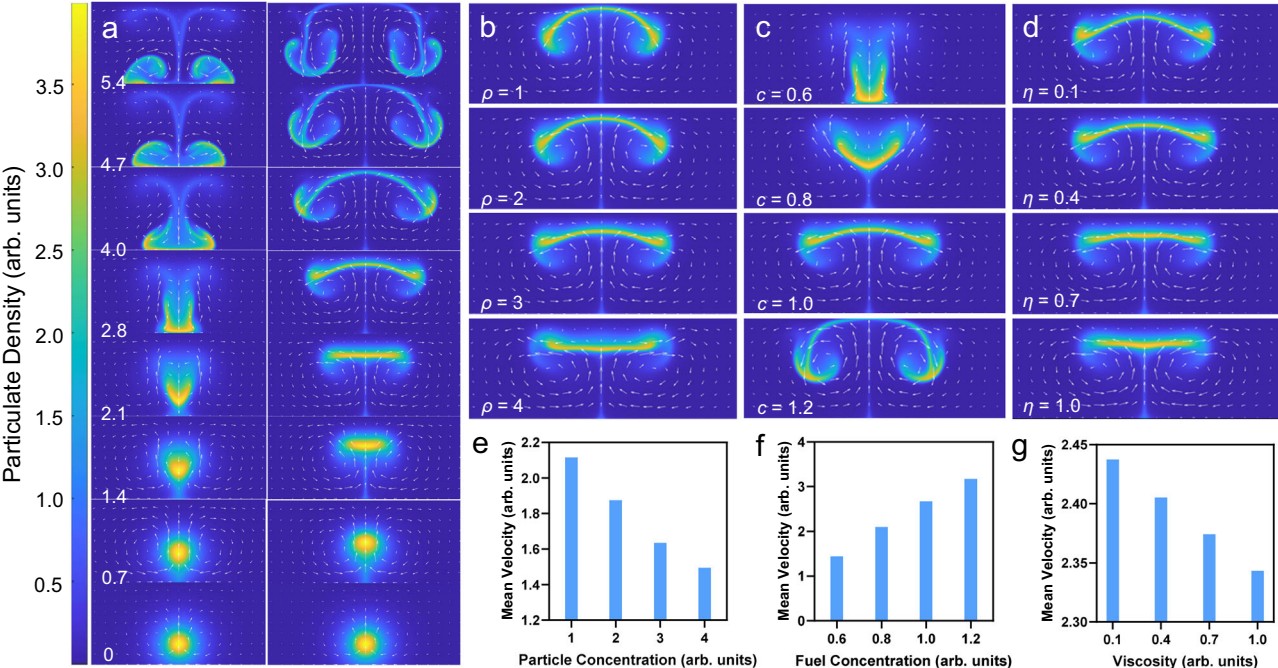

**Fig. 5 | Computational modelling shows UrNMs swarms move upward under different conditions. a** Simulation snapshots show two representative collective behaviours of UrNMs swarms in low fuel concentration (left) and in viscous fuel (right). The colour bar depicts the particulate density $\rho$, and white arrows display the fluid flow velocity. Snapshots of videos at dimensionless time 2.8 under different conditions: **b** various concentrations of particulate ($\rho = 1$ - 4), **c** particulate in different fuel concentrations ($c = 0.6$ - 1.2), and **d** particulate in fuel with different viscosity ($\eta = 0.6$ - 1.2). The domain of integration size $160 \times 40$ dimensionless units, number of grid points $1024 \times 256$. Panels **e**–**g** correspond to mean velocity quantification during the upward motion. The dimensionless unit of length corresponds to 0.1 mm, and the dimensional unit of time corresponds to 1·10 seconds of the experiment depending on the parameter choice.

where $z_0\rho\epsilon c$ is the volume buoyancy force due to gas generation, $z_0$ is the unit vector in the $z$-direction, the gas is produced due to the reaction between fuel $c$ and particulate $\rho$ with the reaction rate $\gamma$. Other parameters: fuel diffusion $D_c$, gravity acceleration $g$, relative particulate/solvent density contrast $\alpha$, $\epsilon$ is the relative buoyancy coefficient that depends on the density of reaction products, and $\kappa_1$ is the normalized drag coefficient. The details of model derivation are presented in Supplementary Note 1. Equations (1)–(3) were solved by the finite difference method using Matlab. We considered a two-dimensional rectangular integration domain (corresponding to the size view) with periodic boundary conditions in the $x$-direction and non-slip conditions in the $z$-direction. The primary difference with models of enzyme-generated solutal buoyancy mechanisms considered in ref. 67. is that the enzyme distribution is not fixed but dynamically updated by the reaction-generated flow.

When buoyancy is not sufficient to counterbalance the gravity of particulates, like in the cases of high concentration of particles and low concentration of fuel, the particulate is not able to rise to the top plane and sink to the bottom after seeding, Fig. 5a, left panel. On the contrary, in the cases of low concentration of particles and high concentration of fuel, particulates rise and spread along the top plane, then descend, experiencing a similar process as in the experiment, Fig. 5a, right panel, and video S7. In simulations, the volume density $\rho$ changes from 1 to 4, chemical fuel $c$ ranges from 0.6-1.2, and kinematic viscosity $\eta$ varies from 0.1-1.0 to simulate different concentrations of particles, fuel, and HA, respectively. In Fig. 5b–d, frames at dimensionless time 2.8 are chosen from computer videos for different parameters. Figure 5b shows that in the same time frame, particulate with smaller density $\rho$ enters the sinking stage, while particulate with larger $\rho$ is still in the ascending or spreading stage, indicating that lighter particulates move faster. This observation agrees with the experimental results and can be further verified by Fig. 5e. The mean velocity of particulate during upward movement decreases with the

increase of density $\rho$. In Fig. 5c, particulate settles to the bottom when chemical fuel concentration $c$ is low ($c = 0.6$). Increasing the $c$ value ($c = 0.8$) triggers particulate's upward movement, yet it settles before reaching the top plane. Only relatively high fuel concentrations force the particulate to go through the three stages, and its upward speed increases with the increase of $c$ value. In Fig. 5f, the gradual increase of the mean particulate velocity with the fuel concentration from simulations agrees with that observed in the experiments. The effect of viscosity is shown in Fig. 5d, g. Particulate in lower viscosity media enters the sinking stage earlier than for higher viscosity. Computational modelling confirms that the increased fuel viscosity slows down the particulate motion.

## Computational modelling of the vertical confinement effects

We performed computational modelling of the effect of vertical confinement on collective behaviour. The details are presented in Supplementary Note 2. The model is derived from Eqs. (1)–(3) by height-averaging using the approach like in ref. 68. The corresponding two-dimensional equations in the $x$-$y$ plane are solved by the quasi-spectral method in the periodic square domain using Matlab.

Parameter $\beta$ is proportional to the reaction rate and parameter $\varepsilon$~$h^2$, where $h$ is the height of the chamber. We adjust the value of these two control parameters to describe the fluid flow slowdown caused by confinement. In Fig. 6a, numerical results show that in vertical confinement, particulate moves dynamically and form aggregates in the centre area of the cell. A similar phenomenon has been observed in experiment, Fig. 4a. However, when the chamber's height is reduced, the fluid flow slows, and the reaction rate decreases. As a result, particulate movement becomes more localized, and the shape formed by a particulate remains almost unchanged within the time durations, as shown in Fig. 6b, c and video S8. Furthermore, there is no significant difference between the swarm dynamics in two highly confined

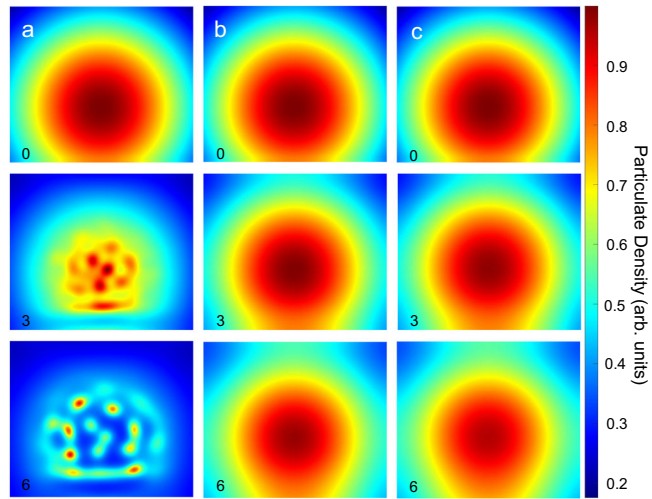

**Fig. 6 | Computational results describing the collective behaviour shaped by confinement.** Snapshots at dimensionless time 0, 3 and 6 in **a** relatively less confined space (large height), for $\beta = 12$, $\varepsilon = 0.05$, the domain of integration size $60 \times 60$ dimensionless units, number of grid points $512 \times 512$; **b** relatively confined space (medium height) for $\beta = 12$, $\varepsilon = 0.03$; and **c** smaller height, for $\beta = 20$, $\varepsilon = 0.01$, respectively. Density $\rho$ is shown in colours.

chambers because fluid convection is inhibited by vertical confinement.

In conclusion, we investigated the collective behaviour of enzymatic nanomotors from the side and from the top. We attribute their collective behaviour to buoyancy-induced convection. When introducing a drop of UrNMs, dispersed in PBS buffer, into a fuel medium (high concentration of urea dissolved in PBS), the UrNMs exhibit directional upward movement due to buoyancy arising from the density difference between the particulate and the fuel medium. UrNMs decompose urea and generate carbon dioxide and ammonia, with the latter dissolving in water, further reducing the particulate density and enhancing its upward movement. When reaching the solid-air interface, UrNMs spread along the interface, form an unstable layer of front, and then sink in the form of finger-like aggregates. The process resembles natural bioconvection in microorganismal suspensions.

Particle concentration, fuel concentration, and viscosity are crucial parameters to control enzymatic collective behaviour. Specifically, increasing particle concentration, decreasing fuel concentration, or increasing viscosity can decrease the density difference between the particulate and the fuel, impeding the initiation of upward movement and subsequent convection. This phenomenon explains the settlement of nanoparticles to the bottom when observed under inverted microscopy. Furthermore, the movement of UrNMs in vertical confinement also serves as a demonstration of buoyancy-induced convection. Confinement hinders fluid convection, indicating that the collective behaviour of enzymatic nanomotors requires vertical spaces to overcome dissipation. While these control factors are essential for understanding collective behaviour, further studies are needed to investigate how to effectively guide swarm dynamics. Possible strategies could involve combining external fields or exploiting collective chemotaxis behaviour.

We performed computational modelling based on the buoyancy-driven convection mechanisms; the results align well with experimental findings. In computational modelling, particulate ascends due to buoyancy, spreads upon reaching the top, and consequently descends because of gravity. Consistent with the experimental observations, an increase in particulate density ($\rho$), a decrease in fuel concentration ($c$), or an increase in fuel viscosity ($\eta$) decreases the mean particulate velocity. Computational modelling also agrees with

experimental observations for particulate moving in vertical confinement. By adjusting the parameters $\beta$ and $\kappa$, corresponding to the reaction rate and the chamber height, respectively, the computational model predicts that vertical confinement shapes the swarms by controlling fluid convection.

The buoyancy-driven convective flow enables the collective movement of enzymatic nanomotors and promotes a more homogeneous particle distribution. In a fuel-rich environment, collective behaviour occurs naturally due to buoyancy and chemical reactions, without requiring external forces. This buoyancy-driven dynamics can be harnessed to design future protocols for large tissue and organ volumes, such as the bladder and joints. It allows overcoming the limitations of current cancer treatments, including sedimentation and poor dispersion in small volumes, thereby facilitating mass transport, accumulation, penetration, and effective diffusivity of individual motors.

## Methods

### Synthesis of MSNPs-NH$_2$

Mesoporous silica nanoparticles (MSNPs) serving as chassis for urease-propelled nanomotors were synthesized by the sol-gel procedure according to our previous report[36]. Briefly, a mixture of TEOA (35 g), Milli-Q water (20 ml), and CTAB (570 mg) was heated to 95 °C under reflux for 30 minutes. TEOS (1.5 ml) was then added dropwise, and the reaction continued for 2 hours. The resulting MSNPs were collected by centrifugation ($2000 \times g$, 5 min) and washed with ethanol, with the process repeated three times. CTAB was removed by refluxing the MSNPs in a methanol (30 ml) and hydrochloric acid (1.8 ml) mixture at 80 °C for 24 hours. Finally, the MSNPs were collected by centrifugation ($2000 \times g$, 5 min), washed in ethanol (three times), and their concentration evaluated by dry weighing.

The surface of MSNPs was then modified for further functionalization. Briefly, 20 mg MSNPs in ethanol 99% (Panreac Applichem cat. no. 131086-1214) and 100 μL 3-aminopropyltriethoxysilane (APTES) 99% (Sigma-Aldrich cat. no. 440140) were mixed and placed in an end-to-end shaker at room temperature for 24 h. The resulting nanoparticles were then collected and washed in ethanol by centrifugation ($2000 \times g$, 5 min) four times to remove residual APTES. The collected MSNPs-NH$_2$ nanoparticles were dried for further use.

### Synthesis of UrNMs and CatNMs

The prepared MSNPs-NH$_2$ nanoparticles (2.5 mg) were resuspended in 1 mL PBS 1× (Thermo Fisher Scientific cat. no. 70011-036) and activated with 100 μL GA 25 wt% (Sigma-Aldrich cat. no. G6257) in an end-to-end shaker for 2.5 hours at room temperature. The activated MSNPs-NH$_2$ were then collected and washed four times in PBS 1× by centrifugation ($2000 \times g$, 5 min), then resuspended in 1 mL PBS 1× with 3 mg urease from *Canavalia ensiformis* (Sigma-Aldrich cat. no. U4002), or in 1 mL PBS 1× with 1 mg catalase from bovine liver (Sigma-Aldrich cat. no. C40). The mixture reacted at room temperature in an end-to-end shaker overnight. The resulting urease-nanomotors (UrNMs)/catalase-nanomotors (CatNMs) were collected and washed thrice in PBS 1× by centrifugation ($2000 \times g$, 5 min). Keep the supernatant of centrifugation for further quantification of the enzyme linkage. Finally, resuspend the collected nanomotors in PBS 1× (0.5 mL) and store them in the fridge at 4 °C for future use.

### DLS measurements of UrNMs

Malvern Nanosizer (Zetasizer Nano ZSP) was used to measure the diffusion coefficient of UrNMs across a range of urea concentrations (0, 50, 100, 150, and 300 mM) and the surface charge of MSNPs, MSNPs-NH$_2$, and UrNMs. We analyzed the diffusion coefficient of UrNMs (20 μg/mL) at each urea concentration and zeta potential values of each type of nanoparticles (20 μg/mL) with three runs per

experiment. Nine measurements per type of particle were performed to obtain statistically relevant data.

## UrNMs characterization

The synthesized MSNPs were characterized by scanning electron microscope (SEM), revealing a uniform particle size distribution centred at 450 nm (Fig. S1a, b). Amino groups were then grafted on the surface of MSNPs, facilitating further modification of urease on the MSNPs surface by linking GA molecules between amino groups. The surface modification process was characterized by zeta potential measurements (Fig. S1c). The introduction of amino groups results in a negative surface charge of MSNPs reversed from $-38.7 \pm 4.61$ mV to a positive surface charge of $27.77 \pm 7.9$ mV. The subsequent linking of GA molecules and urease is confirmed by particle surface charge changes due to the presence of abundant aldehyde groups and carboxyl groups, with negative surface charge reverses to $-14.5 \pm 9.13$ mV and $-9.02 \pm 4.34$ mV for GA molecule and urease, respectively. DLS measurement indicates that the prepared UrNMs show an enhanced diffusion coefficient in elevated urea concentrations (Fig. S1d).

## Optical video recording and nanoparticle tracking

The collective behaviour of UrNMs in a vertically confined space was recorded using a Leica DMi8 microscope equipped with a high-speed cooled charge-coupled device (CCD) camera from Hamamatsu and a $2.5\times$ objective lens. Two pieces of cover glasses were separated by spacers (Silicone isolators from Grace Bio-Labs) with varying heights: 1.6 mm, 0.5 mm, and 0.25 mm. The confined space was then filled either with PBS or with a 300 mM urea solution in PBS and positioned under the microscope. A drop of UrNMs or MSNPs ($3\,\mu$L) was added to the liquid-filled chamber, and videos (25 fps, 2 min) were recorded. Optical videos from the side view were recorded using either a digital camera (Thorlabs, DCC1240M-GL/Thorlabs, CS165CU) equipped with a lens (FUJINON, HF35HA-1S) or a Leica DFC3000G camera equipped with a $10\times/0.3$ objective lens. A $22 \times 1.6 \times 8$ mm (length × width × height) chamber was prepared by separating two pieces of cover glasses with spacers. A drop of UrNMs or MSNPs ($3\,\mu$L) was added to the liquid-filled chamber and videos (15 fps, 2 min) were recorded. Then these videos were analyzed using a home-designed programme in Python[27].

The motion profiles of urease-powered nanomotors were analyzed in a 50 mM urea solution with varying ionic strengths, achieved by adding different concentrations of NaCl (0.05, 0.5, 5, 25, 50 mM). Videos were recorded at 25 fps for 30 seconds using an inverted optical microscope (Leica DMi8) equipped with a Hamamatsu digital camera (C11440) and a $63\times$ water immersion objective. For each experiment, a drop of urea solution with the specified NaCl concentration was placed on a glass slide, followed by the addition of $3\,\mu$L of UrNMs. The glass slide was then covered with a coverslip. The acquired videos were analyzed using a custom-designed Python software. The mean square displacement (MSD) can be calculated from the extracted trajectories by the equation MSD($\Delta$t) = $4D\Delta$t, where $D$ is the diffusion coefficient.

## Bicinchoninic acid (BCA) assay

The amount of urease or catalase linked onto the MSNPs surface was quantified by BCA analysis (Thermo Fisher Scientific cat. no. 23227), table S2. Bovine serum albumin (BSA) was used as the standard for quantifying the concentration of protein concentrations. First, a series of BSA concentrations 2000, 1500, 1000, 750, 500, 250, 125, 25, 0 $\mu$g/mL were prepared, and BSA solutions and the as-prepared supernatant of UrNMs or CatNMs samples were added separately into a 96-multiwell plate, 25 $\mu$L for each well. Then 200 $\mu$L working reagent (light sensitive), made with 50 parts of reagent A and 1 part of reagent B, was added to each well that has been used, either with the BSA standards or the sample. Next, shake the plate for 30 s to mix the solutions, and incubate the reaction for 30 min at 37 °C. Afterward, the absorbance of both BSA solutions and the samples after the reaction was measured at a wavelength of 562 nm. By comparing the protein quantity remaining in the supernatant to the initial amount of protein added and the standard concentrations of BSA, the amount of attached enzyme can be quantified.

## Enzymatic activity measurement

Urease activity was detected before and after being linked on the surface of MSNPs and was compared in PBS and acetate buffer. 0.025 mM Phenol red (Sigma-Aldrich cat. no. 114529) was added to different concentrations of urea solutions (0, 50, 100, 200, and 300 mM). The solvent of these solutions could be either PBS or acetate buffer. 2 $\mu$L PBS solution or UrNMs or urease (130 $\mu$g/mL) were added in a 96-well plate separately, followed by the addition of urea solutions. The 96-well plate was immediately placed in a multimode microplate reader (BioTek Synergy HTX). The absorbance changes of phenol red were measured in real time at a wavelength of 560 nm for 60 min for measurements in PBS buffer and 100 min for acetate buffer. The incubation was performed at room temperature with a minimum measurement interval of 30 seconds, and the sample was orbitally shaken at a minimal frequency.

## Estimation of specific enzymatic activity

The specific enzymatic activity was determined by calculating the slope of the enzymatic activity curve. According to the Beer-Lambert law,

$$A = klc$$

where $A$ is the absorbance, $k$ is the molar attenuation coefficient of phenol red, and $l$ is the path length of 0.5 cm, the concentration changes $c$ of phenol red per minute were computed. Subsequently, the specific enzymatic activity was derived based on the quantity of enzyme utilized.

## Gas detection

The generated $CO_2$ and $NH_3$ were identified using a gas detector (Dräger X-am 7000). In a glass bottle filled with urea dissolved in PBS (10 mL, 200 mM) or acetate buffer (10 mL, 200 mM), UrNMs (2.5 mg) were added, and the caps were securely fastened to prevent gas release. After 30 min, the bottle caps were removed, and the probe of the gas detector was placed over the solutions to record the generated gases.

## Videos analysis

To investigate the dynamics of swarms over time, the recorded videos were analyzed by pixel intensity distribution and density maps. For pixel intensity distribution, snapshots of videos were captured at 12-second intervals. Subsequently, a region of interest (ROI) measuring 300 pixels by 300 pixels was selected, and the pixel intensity distribution within the ROI was analyzed using ImageJ software. To perform density map analysis, the videos were initially processed to remove the background using ImageJ software. Then 40-second segments were extracted from these videos. The cumulative pixel intensity of these segments, consisting of 1000 frames each, was computed and visualized using the turbo colormap.

## Particle image velocimetry (PIV)

The PIV of recorded videos was conducted by a custom Python code based on the OpenPIV library. The consecutive frames of videos within desired time intervals were extracted and then loaded into the code OpenPIV, with an interrogation window size of $32 \times 32$ pixels (width×height), an overlap of $16 \times 16$ pixels (horizontal×vertical), and a frame rate of 3.33 fps. The results were then reloaded into the Python code to adjust the arrow size and display particle velocities in colour bars.

## Data availability

All data supporting the findings are available within the article and the Supplementary Information. The raw data generated in this study have been deposited in the Figshare database https://doi.org/10.6084/m9.figshare.27134523.v1[69].

## Code availability

The custom scripts used for computational analysis in this study are available on GitHub at https://github.com/SC357/Convective_Dynamics[70]. The code is provided under the MIT License. Additional information can be found in the repository's README file and can also be requested from the corresponding author.

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

## Acknowledgements

The research leading to these results has received funding from the grants PID2021-128417OB-I00 and PDC2022-133753-I00 funded by MCIN/AEI/ 10.13039/501100011033 and, by "ERDF A way of making Europe" and European Union Next Generation EU, (Bots4BB and BOJOS projects) (S.S.). This project has also received funding from the European Research Council (ERC) under the European Union's Horizon 2020 research and innovation program (grant agreement No 866348, iNanoSwarms) (S.S.). The IBEC team wishes to thank the CERCA programme of the Generalitat de Catalunya, the Secretaria d'Universitats i Recerca del Departament d'Empresa i Coneixement de la Generalitat de Catalunya through the project 2021 SGR 01606, and the "Centro de Excelencia Severo Ochoa", funded by Agencia Estatal de Investigación (CEX2018-000789-S). We thank Santiago Marco Colás and Eduardo Caballero Saldivar for their technical support. S.S and S.C. acknowledge the Predoctoral AGAUR-FI Joan Oró grant (2023 FI-1 00654) funded by "Secretaria d'Universitats i Recerca del Departament de Recerca i Universitats de la Generalitat de Catalunya" and by European Social Fund Plus. The research of I.S.A. was supported by the NSF award PHY-2140010.

## Author contributions

S.C. synthesized and characterized the nanomotors, performed the motion and velocity analysis, analyzed the data, and wrote the paper. X.P. developed Python scripts for the upward velocity analysis and developed Matlab scripts for the computational modelling. A.C.B. designed the side view setup and contributed to the simulation work. F.L. analyzed the centre of mass tracking and contributed to the data processing. I.S.A. conceived the initial idea, developed the computational modelling, and edited the paper. S.S. conceived the initial idea, supervised the project, and edited the paper.

## Competing interests

The authors declare no competing interests.
