## [Transparent Peer Review file · Nature Communications]

Collective Buoyancy-driven Dynamics in Swarming Enzymatic Nanomotors

Corresponding Author: Professor Samuel Sánchez

Version 0:

Reviewer comments:

Reviewer #1

(Remarks to the Author)

Manuscript ID: NCOMMS-24-12498

Manuscript title: Convective Dynamics of Swarming Enzymatic Nanomotors

Authors: Shuqin Chen, Xander Peetroons, Anna C. Bakenecker, Florencia Lezcano, Igor S Aranson, Samuel Sánchez

Reviewer's report: In this work, Shuqin Chen et. al. report research on the swarming of enzymatic nanomotors made from mesoporous silica nanoparticles with urease attached. The introduction covers a wide range of topics, including swarming behavior, synthetic systems, micro/nanomotors, and bioconvection, demonstrating the breadth of the research. It effectively demonstrates the complexity and importance of collective behavior in organisms with compelling examples. It also highlights the potential biomedical applications of enzymatic nanomotors, emphasizing their role in targeted drug delivery and cancer therapy.

The study systematically describes enzymatic nanomotor swarms displaying gravitactic behavior and collective movement in three dimensions (3D). The authors propose a model likening the swarm to light particulates in a denser fluid, moving upwards and creating convective flows within a closed fuel-filled environment. This phenomenon bears resemblance to bioconvection and provides insights into the mechanisms underlying swarming behavior in enzymatic nanomotors.

In my opinion, through this work, the authors propose an interesting idea for biomedical applications of enzymatic nanomotors. However, the manuscript is not technically advanced enough and necessary improvement based on the following suggestions will help attract a broad readership.

The introduction lacks clarity and engagement due to convoluted structure, too much technical terminology, and a lack of clear definitions. It fails to explain how UrNMs exhibit emergent properties or the benefits of group flow. The aim of the research is unclear, and biomedical applications are mentioned briefly without explanation or evidence. Overall, the introduction lacks organization and relevance, making it hard for readers to understand the research's significance.

It was found that there are reports from the same group focusing on similar nanomotors based on urease functionalized mesoporous silica in the fuel of Urea.

1. One study reported in ACS Nano,

'Enzyme-Powered Gated Mesoporous Silica Nanomotors for On-Command Intracellular Payload Delivery' Antoni Llopis-Lorente et al, 2019, 13, 12171–12183

The study introduces enzyme (urease)-powered nanomotors based on mesoporous silica with cargo, capped for controlled release at acidic pH, showing improved delivery in HeLa cells, hinting at their promise for responsive drug transport.

2. Also, in another study reported in the Science Robotics,

'Swarming behavior and in vivo monitoring of enzymatic nanomotors within the bladder' Hortelao et al., 6, eabd2823 (2021)
The study also reveals urease-powered nanomotors' collective behavior tracked by positron emission tomography (PET),

showcasing improved fluid mixing and migration. These nanomotors were administered directly into the bladder of mice for the advanced study in clinical theranostic use.

Both these studies and the current manuscript (NCOMMS-24-12498) have used similar mesoporous silica-based enzymatic (urease) nanomotors. The reported studies have technical advantages and further applications in practical use. However, the current work is systematic or phenomenological study that lacks technological advances as well as further applications. It seems that the nanomotor's flow is like diffusive behavior or flow and the efficient control parameters are also not described properly.

Below is a list of some of my observations that must be addressed before recommending the manuscript for publication in Nature Communications.

1. In Figure 3a, this graph shows that when adding varied amounts of UrNMs into urea dissolved in PBS buffer, the pH of the solution then rises, which means that some or all of the ammonia produced by the urease catalysis reaction is dissolved. Which is different from the conclusion described in lines 192 to 195. And there is no bubble observed in the first cuvette of video S3. This also indicates that most of the gas produced is dissolved in the PBS buffer.
2. Page 3, lines 88: UrNMs are dispersed in phosphate buffer saline (PBS) buffer as shown in Fig.1. However, in video S3, more bubbles are generated in the acetate buffer compared to PBS buffer. This means UrNMs would have swarm behavior in the acetate buffer.
3. Enzyme-powered MNMs utilize enzymatic catalysis for directional mobility. Studying particle and fuel concentration uncovers insights into collective movement. In fuel-rich environments, nanomotors move against gravity. However, exploring lateral flow and other directions is vital for comprehensive understanding and broader applications.
4. In Figure 2c, both UrNM and MSNP velocities are plotted against UrNM concentration. Were UrNM and MSNP considered separately? Specifically, does UrNM refer to a composite of Ureas and mesoporous silica? ii. Despite MPNPs exhibiting high velocity at low concentration, how can this phenomenon be explained?
5. The velocity of UrNM and MSNP against UrNM concentration is confusing, as it implies that MSNP does not contain any UrNM. Is it suggesting that both UrNM and MSNP originate from the same complex depicted in Figure 2.c?

In addition, the authors need to consider the following minor issues:

1. Page 1, lines 35: There is an extra space between high and efficiency.
2. Page 2, lines 62: nanomotors'
3. UrNM and UrNMs: Are these different or are they just singular and plural, it's confusing in some cases. Please clarify.
4. Page 6, lines 182: Here describe the contents of video S3. However, in the video, it is observed that UrNMs and Urease are added to the cuvettes at different moments. While turbidity is observed in the first two containers. These phenomena are not described in the article.
5. Page 6, lines 197: Verification of the products of UrNMs catalysis reaction accelerate (accelerates) directional movement.

Reviewer #2

(Remarks to the Author)

Report: NCOMMS-24-12498: "Convective Dynamics of Swarming Enzymatic Nanomotors"

The authors present an experimental combined with theoretical modeling study of the dynamics of enzymatic nanomotors. The article presents intriguing findings, yet the main findings outlined in the introduction lack novelty and might depend on the specific model configuration chosen. While timely, the study's novelty is challenging to assess given recent similar studies in the literature. For instance, characterizing enzymatic nanomotors' upward directional motion as a "swarm" may not be entirely accurate, as swarming typically involves collective swirling or whirling motions observed in bacterial swarms or larger-scale swarms. The analysis appears to focus primarily on the interplay of gravity and buoyancy dynamics without delving into collective behavior. A deeper analysis of the dynamics and terminology usage may be necessary for a comprehensive understanding. I would encourage the authors to present a comparison of their findings with existing literature.

Further, the following major points need to be considered before publication in any journal:

1. While the exploration of particle concentration, fuel concentration, viscosity, and vertical confinement is valuable, I propose conducting a control experiment to further elucidate the system's behavior. By confining the system to two dimensions where convection is not feasible, we can observe whether directed motion persists in the absence of vertical movement. This additional experiment would provide insight into the underlying mechanisms driving the nanomotor swarm's motion and help validate the findings under different conditions.
2. It's indeed a pertinent question. If the authors classify the observed phenomenon as "swarming," the absence of vortex motion within the swarms raises questions about the accuracy of that classification. Vortex motion, characteristic of swarming behavior, typically involves collective swirling or whirling movements within the swarm. The lack of such motion suggests that the dynamics observed may not align entirely with traditional swarming behavior. Clarifying this discrepancy could lead to a better understanding of the true nature of the observed phenomenon and its underlying mechanisms.
3. The distinction between active and passive swarms, as mentioned by the author in the control experiments, lacks clarity. While both types exhibit directional motion, it remains unclear why one is labeled as active and the other as passive.

Reviewer #4

(Remarks to the Author)

This manuscript provides an investigation into the swarming behavior of enzymatic nanomotors based on urease (UrNMs) driven by buoyancy-induced convection. UrNMs exhibit upward movement in fuel mediums due to density differences, which are influenced by particle concentration, fuel concentration, and viscosity. Settlement behaviors and vertical confinement experiments further validate these phenomena. Computational modeling aligns with the experimental findings, demonstrating how adjusting parameters can control swarm behavior. These insights contribute to the understanding of collective motion. In summary, the manuscript provides an innovative foundation for understanding the swarming behavior and underlying mechanisms of urease powered nanomotors. The result is logical and reasonable. However, to ensure compliance with the publication standards of Nature Communications, the manuscript is suggested to make some modifications as follows:

1. The manuscript mentions that "Our findings are crucial for the potential biomedical applications of enzymatic nanomotor swarms, ranging from enhanced diffusion in bio-fluids and targeted delivery to high- efficiency cancer therapy". What are the potential applications of urease-based nanomotors, and what is the current status of their development? Please provide a concise explanation about their applications and technological advancements.
2. Swarming behavior encompasses collective motion exhibited by self-propelled entities, often characterized by coordinated interactions and emergent patterns. In the context of enzymatic nanomotors, swarming behavior refers to the tendency of these particles to aggregate or form spatially organized groups. Understanding the underlying mechanisms driving this clustering phenomenon, such as inter-particle interactions, environmental cues, or external stimuli, is crucial for deciphering the dynamics and functionalities of these swarms. The commonly used strategies for the assembly and control of swarms are divided into four kinds in the manuscript. One of them is "(3) combining biological microswimmers, such as sperm cells and algae, into artificial moieties as a hybrid integration." Could this phenomenon be classified as a form of swarming behavior, and if so, what specific clustering behaviors do these entities demonstrate?
3. In my mind, there are a series of articles concerning convection-induced swarming motors from, such as the group of Prof. Guan and others. It is suggested to make a comparative description about them. In this way, readers may know more its whole story.
4. Blood contains water, electrolytes, proteins, hormones, cells, and waste products like urea and creatinine, which has a density of 1.050 to 1.065 g/mL and a viscosity of 3.4 to 5.3 mPa·s at 37°C. On the other hand, urine consists of water, urea, creatinine, uric acid, electrolytes, and waste products. Its density ranges from 1.003 to 1.030 g/mL, and viscosity is depending on solute concentration. Could the presence of complex components in blood or urine exert an influence on the swarming behaviors of nanomotors? Exploring the interactions between nanomotor clusters and the intricate molecular compositions of biological fluids like blood and urine would provide valuable insights into the feasibility and efficacy of nanomotor-based applications in biomedical contexts. Factors such as protein binding, pH variations, and electrolyte concentrations within these biological fluids may modulate the behavior and performance of nanomotor clusters, necessitating a thorough investigation to ascertain their functional capabilities and limitations in complex physiological environments.
5. The role of temperature is paramount in enzymatic reactions, with significant implications for the movement dynamics of both passive particles and nanomotors. A comparative analysis of the movement behaviors of passive particles and nanomotors under physiological temperature conditions is recommended. This analysis would shed light on how temperature influences the kinetic properties, diffusion rates, and overall mobility of these particles, elucidating potential advantages or challenges faced by nanomotors in biological environments. Furthermore, exploring the temperature-dependent responses of nanomotors in relation to enzymatic activity and propulsion mechanisms can provide valuable insights into their performance and functionality in biomedical applications. Integrating temperature considerations into the study of nanomotor behavior contributes to a comprehensive understanding of their behavior under realistic physiological conditions, guiding the optimization of nanomotor design for enhanced efficacy and versatility in biomedical settings.
6. Is the mechanism proposed in this paper universally applicable to a wide range of enzyme-driven nanomotors developed thus far, not restricted to urease-based systems? It is imperative to investigate the generalizability of this mechanism across various types of enzyme-driven nanomotors to elucidate fundamental principles governing their propulsion and collective behaviors. This inquiry not only contributes to a deeper understanding of nanomotor design and optimization strategies but also sheds light on potential commonalities or distinctive characteristics among different enzyme-driven systems, thereby advancing the field of nanotechnology.
7. In practical scenarios such as the blood circulatory system or bladder, nanomotors frequently encounter fluidic or irregularly shaped operating environments. Therefore, it is essential to simulate and analyze the clustering behavior of these motors within fluidic or flexible materials. Such simulations not only aid in understanding the dynamic interactions and spatial organization of nanomotor clusters but also inform the design and optimization of these nanoscale systems for enhanced performance and functionality in complex biological environments.

Version 1:

Reviewer comments:

Reviewer #1

(Remarks to the Author)

Major comments; Thanks to the authors for the revisions made to the introduction. After thorough review, I am not convinced

using the swarm terminology in the study. Our primary concern lies in interpreting and characterizing the observed phenomena as "swarming." Swarming is typically characterized by several key features, including dynamic interactions among agents, self-organization, and emergent properties that result from local interactions rather than global control. In biological systems, swarming behavior often involves complex coordination and communication between individual agents, leading to collective decision-making, adaptive responses to environmental changes, and sophisticated task execution. In this study, the described behavior of enzymatic nanomotors appears to be driven primarily by buoyancy-induced convection rather than dynamic interactions among individual nanomotors. While the observed collective movement and convective flows are indeed intriguing, they seem to result from the physical properties of the system rather than emergent behaviors arising from interactions among the nanomotors themselves. The distinction between convection-driven aggregation and true swarming behavior is crucial, as the latter implies a higher level of complexity and interaction.

The remaining concerns are as follows;

To the author's response 1: I appreciate your effort to address the mismatch between the experimental data and the manuscript's conclusions. Your revised conclusion provides a clearer understanding by stating that urease catalysis produces both dissolved NH₃ and CO₂ in PBS buffer, and NH₃ and CO₂ gas in acetate buffer. This clarification helps reconcile the experimental results with the conclusions drawn.

To the author's response 2: I appreciate the clarification regarding the bubble generation differences between the acetate and PBS buffers and the subsequent investigation into the swarming behavior of UrNMs in acetate buffer. Your response and the new data provided, including the time-lapse images and velocity analysis, offer valuable insights into the effects of different buffers on the behavior of UrNMs. However, the experiments demonstrate that UrNMs exhibit collective movement in both PBS and acetate buffers, the fundamental issue remains that the observed behaviors do not strongly distinguish between different types of collective phenomena, such as true swarming versus buoyancy-driven motion. For improvement, I suggest: clarifying how the observed behaviors in different buffers can be distinguished from other types of collective phenomena; providing detailed mechanistic insights into why different buffers lead to variations in bubble generation and nanomotor behavior; and conducting additional experiments to explore other factors influencing swarming behavior and verify whether the observed phenomena in acetate buffer truly represent swarming.

To the author's response 3: Thank you for your detailed response and for updating the figures to include additional data on the trajectories of the enzymatic nanomotors. I also appreciate the effort you have put into enhancing the visual representation of your work. However, the newly added content, specifically the images and trajectory data, does not significantly contribute to a deeper understanding of the underlying mechanisms of the observed behavior. The updated figures primarily illustrate the clockwise and counterclockwise flow patterns of the enzymatic nanomotors. While these visuals effectively demonstrate the flow directions, they do not offer new insights into the mechanisms driving these movements. The observed phenomena appear to be a result of fluid dynamics rather than an indication of different underlying mechanisms or behaviors.

Reviewer #2

(Remarks to the Author)

Report: NCOMMS-24-12498: "Convective Dynamics of Swarming Enzymatic Nanomotors"

The authors have successfully addressed all my queries, and the revised manuscript is significantly improved with respect to the scientific relevance and clarity of presentation. The manuscript is now suitable for publication.

Reviewer #4

(Remarks to the Author)

The authors have responded to all the comments and suggestions. I am satisfied by the most of the answers. Thus, the manuscript has been greatly improved. At the same time, I have still several issues to be addressed in the following.

1. The authors did not discuss propulsion mechanism in the revised manuscript. In fact, urease-powered motors may be propelled with different propulsion mechanisms, such as ionic diffusiophoresis and microbubble recoils depending on the urea concentration and the motor structure. Will whether the propulsion mechanism affect the swarming behavior? As the motor in this work was about 500 nm in size, similar to that reported in (ACS Nano, 2023, 17, 6023–6035), the propulsion is possible to be dominated by ionic diffusiophoresis.
2. The ureases were modified on the whole surface of the mSiO₂. In this case, how to simulate the distribution of products or fluid around these nanomotors? What will happen if the nanomotor becomes a Janus structure? Additionally, does the size of motors affect this phenomenon?
3. Some of the figures, such as Fig. 4, Fig. 6, and Figs. S13-S21, show very poor quality. They are difficult to read clearly. The author should optimize these images.
4. The biomedical applications are still briefly mentioned, but without explanation or evidence. This will limit the significance of the work.

Version 2:

Reviewer comments:

Reviewer #1

(Remarks to the Author)

The authors have satisfactorily addressed all my concerns, and the revised manuscript has significantly improved in terms of both scientific relevance and clarity.

Reviewer #4

(Remarks to the Author)

I fully understand that the authors have seriously replied to all the reviewer's comments, despite the lack of corresponding experimental data. Of course, if there are data supported, it will be more conducive to improving the quality of this manuscript. In addition, I would feel more gained if the author could clarify the possible specific contribution of the formation mechanism of swarming UrNMs mentioned to the future development of swarming micro/nanomotors in biomedicine applications, rather than simply enumerating the examples in biomedicine applications.

Point-to-point Responses to Comments

Reviewer #1:

Reviewer's report: "In this work, Shuqin Chen et. al. report research on the swarming of enzymatic nanomotors made from mesoporous silica nanoparticles with urease attached. The introduction covers a wide range of topics, including swarming behavior, synthetic systems, micro/nanomotors, and bioconvection, demonstrating the breadth of the research. It effectively demonstrates the complexity and importance of collective behavior in organisms with compelling examples. It also highlights the potential biomedical applications of enzymatic nanomotors, emphasizing their role in targeted drug delivery and cancer therapy.

The study systematically describes enzymatic nanomotor swarms displaying gravitactic behavior and collective movement in three dimensions (3D). The authors propose a model likening the swarm to light particulates in a denser fluid, moving upwards and creating convective flows within a closed fuel-filled environment. This phenomenon bears resemblance to bioconvection and provides insights into the mechanisms underlying swarming behavior in enzymatic nanomotors.

In my opinion, through this work, the authors propose an interesting idea for biomedical applications of enzymatic nanomotors. However, the manuscript is not technically advanced enough and necessary improvement based on the following suggestions will help attract a broad readership.

The introduction lacks clarity and engagement due to convoluted structure, too much technical terminology, and a lack of clear definitions. It fails to explain how UrNMs exhibit emergent properties or the benefits of group flow. The aim of the research is unclear, and biomedical applications are mentioned briefly without explanation or evidence. Overall, the introduction lacks organization and relevance, making it hard for readers to understand the research's significance."

Author's response: We thank the reviewer for this suggestion about the Introduction. To effectively underscore the significance of this work, we have revised the introduction section based on the reviewer's suggestion as follows:

"Swarming behaviour is widespread in nature. While individual units of a group obey simple rules, they present complex and intriguing collective behaviour when assembling into highly ordered structures¹. Living organisms use distributed or swarm intelligence to accomplish sophisticated tasks to survive. Examples range from collective cell migration², honeybees adapting to repeated shaking to maintain mechanical stability of the swarm³, to emperor penguins packing in a huddle in a highly coordinated manner to survive cold winter⁴. Multiple

synthetic swarming systems have been developed with inspiration from nature including: (1) applying one or multiple external forces, such as magnetic fields⁵⁻⁸, light^{9,10}, ultrasound^{11,12}, electric fields^{13,14}, (2) utilizing chemicals as signals¹⁵⁻¹⁷, (3) combining biological microswimmers, such as sperm cells and algae, into artificial moieties as a hybrid integration¹⁸⁻²⁰, (4) exploiting DNA base-pair interaction^{21,22}. These well-designed swarms show many advantages compared to single-unit functionalities, like enhanced coverage and fluid mixing, intelligent multitasking, collective chemotaxis and perception, and environmental adaptation.

Micro/nanomotors (MNMs) are synthetic active devices achieving self-propulsion through converting various types of energy into mechanical motion^{23,24}. Earlier works on enzyme-powered MNMs have demonstrated the motion of single particles²⁵⁻²⁷ and small clusters²⁸, as well as proof-of-concept studies in drug delivery²⁹⁻³³ and sensing^{34,35}. Nonetheless, recent reports have shifted focus to the collective motion of these particles. Recently, Hortelao et al.³⁶ reported the emergent swarming behaviour of enzymatic nanomotors. The urease-powered nanomotors show collective migration in urea, demonstrating the ability to swim across complex paths compared to the inactive nanomotors. Furthermore, the active collective dynamics, combined with advanced imaging technologies, position them as promising tools in the field of biomedicine. For example, swarms of radio-labelled nanobots have shown an eightfold increase in tumour penetration and approximately a 90% reduction in tumour size during radionuclide therapy³⁷. Swarms of catalase-powered nanobots overcome and disrupt mucus layer, resulting in a 60-fold increase in mucus barrier penetration, through in vitro and ex vivo validation³⁸. Hyaluronidase and urease nanomotor swarms work synergistically for enhanced diffusion in viscous media, such as synovial fluid, paving the way for treating joint injuries³⁹. Similarly, collagenase-powered MNMs^{40,41} and urease-powered iron oxide nanomotor swarms⁴² were exploited to disrupt collagen fibers, serving as a model of the extracellular medium. This disruption facilitates cell spheroids penetration and enhances the delivery efficiency of a second swarm of nanomotors by 10-fold.

Although enzyme-powered MNMs have primarily demonstrated their potential in biomedical applications, the mechanisms underlying the emergent swarming behaviour remain to be clarified. Inspired by nature, the intriguing collective phenomenon bears a resemblance to bioconvection. Bioconvection is a self-organized and self-sustained vortex motion that arises naturally in suspensions of microorganisms⁴³. It visually resembles the Rayleigh–Bénard convection in fluid heated from below⁴⁴. The bioconvection emerges due to the unstable density gradients resulting from the accumulation of buoyant microorganisms⁴⁵. Each microorganism plays a pivotal role in driving accumulation and fluid flow. Certain gravitactic algae or aerotactic bacteria exhibit upward swimming. In the presence of an

upper surface, they form a thin boundary layer of microorganism-rich heavier fluid, which becomes unstable, leading to the formation of falling plumes⁴⁶.

In synthetic MNMs, buoyancy-driven convection has been employed for directional motility and cargo delivery. One approach utilizes incident light to generate convective flow through the photothermal effect. This convective flow can drive TiO₂ micromotors to aggregate and form clusters^{47,48}, or enable magnetic colloidal collectives to drift using fluidic currents⁴⁹. Another approach involves enzymes fixed on a surface, which catalyze fuels, inducing density variations between the reactants and products of chemical reactions. For instance, urease-attached macroscale sheets exhibit clockwise or counter-clockwise rotation via a solutal buoyancy mechanism arising from enzyme pumps⁵⁰. Additionally, urease attached to a lipid membrane induces fluid flow and the directional transport of tracer particles⁵¹. These studies underscore the significant implications of convective flow on the collective movement of micro/nanoparticles.

Here, the buoyancy-driven convective dynamics of swarming enzymatic nanomotors arises from the density difference between the product-rich particulate and the denser environment. We attach enzymes onto the surface of nanoparticles, which move dynamically with the fluid flow. We describe the emergent swarming behaviour of enzymatic nanomotor in three dimensions (3D) to unveil the underlying mechanisms. We model a swarm as light particulates immersed in a denser fluid environment. The particulate swarm moves upward, creating a convective flow in a closed fuel-filled space. Control factors for directional and collective mobility of enzymatic nanomotors, such as particle and fuel concentration and media viscosity, are investigated. In 3D space, convective flows develop complex patterns such as vortices. To further validate our findings, we vertically confine the system to two dimensions, where convective flow is constrained to a plane, inhibiting the possible flow configurations. The convective dynamics resemble bioconvection and provide insights into the mechanisms of swarming behaviour observed in enzymatic nanomotors.”

Reviewer’s report: “It was found that there are reports from the same group focusing on similar nanomotors based on urease functionalized mesoporous silica in the fuel of Urea.

1. One study reported in ACS Nano,

‘Enzyme-Powered Gated Mesoporous Silica Nanomotors for On-Command Intracellular Payload Delivery’ Antoni Llopis-Lorente et al, 2019, 13, 12171–12183

The study introduces enzyme (urease)-powered nanomotors based on mesoporous silica with cargo, capped for controlled release at acidic pH, showing improved delivery in HeLa cells, hinting at their promise for responsive drug transport.

2. Also, in another study reported in the Science Robotics,

'Swarming behavior and in vivo monitoring of enzymatic nanomotors within the bladder' Hortelao et al., 6, eabd2823 (2021)

The study also reveals urease-powered nanomotors' collective behavior tracked by positron emission tomography (PET), showcasing improved fluid mixing and migration. These nanomotors were administered directly into the bladder of mice for the advanced study in clinical theranostic use.

Both these studies and the current manuscript (NCOMMS-24-12498) have used similar mesoporous silica-based enzymatic (urease) nanomotors. The reported studies have technical advantages and further applications in practical use. However, the current work is systematic or phenomenological study that lacks technological advances as well as further applications. It seems that the nanomotor's flow is like diffusive behavior or flow and the efficient control parameters are also not described properly."

Author's response: We appreciate the reviewer's feedback. This work utilizes the same chassis and urease as our earlier publications. The difference is that previous publications focused primarily on urease-powered nanomotors at the single-particle level regarding their biomedical potential, as highlighted in the first publication mentioned by the reviewer (Llopis et al. ACS Nano 2019). Then, the group reported the emergent swarming behavior of the urease-powered nanomotors, which paved the way for their applications in theranostics in the bladder of mice (Hortelao et al. Sci. Robot. 2021 and Simo & Serra et al. Nat. Nanotech. 2024). However, while previous papers on urease-powered nanomotors' collective behavior primarily emphasize their biomedical potential, they lack in-depth explanations of the fundamental mechanisms of motion, which are crucial for further investigations. This work mainly investigates the basic mechanisms, serving as a bridge connecting emergent swarming behavior with its potential biomedical applications. We have added these earlier publications in the introduction as indicated in the previous comment above.

Fluid flow has a significant impact on the enzymatic nanomotor's collective motion, and we found that the buoyancy-driven convective flow is the primary driving force. Particle concentration, fuel concentration, and viscosity are three essential parameters that control convective dynamics. However, developing effective control techniques for the nanomotors' collective motion was not a goal of this work. We clarified these limitations in the conclusion section as "*While these control factors are essential for understanding swarming behavior, further studies are needed to investigate how to effectively guide swarm dynamics. Possible strategies could involve combining external fields or exploiting collective chemotaxis behavior.*"

Reviewer: "Below is a list of some of my observations that must be addressed before recommending the manuscript for publication in Nature Communications.

1. In Figure 3a, this graph shows that when adding varied amounts of UrNMs into urea dissolved in PBS buffer, the pH of the solution then rises, which means that some or all of the ammonia produced by the urease catalysis reaction is dissolved. Which is different from the conclusion described in lines 192 to 195. And there is no bubble observed in the first cuvette of video S3. This also indicates that most of the gas produced is dissolved in the PBS buffer.”

Author’s response 1: We thank the reviewer for pointing out this mismatch between experiments and explanations in the manuscript. In the text, we state that *“The above results indicate that urease catalysis reaction produces NH_3 and dissolved CO_2 in PBS buffer, and NH_3 and CO_2 gas in acetate buffer, which are the main reasons that cause accelerated directional movement of active UrNM swarms in urea.”* It’s true that most of the gas produced is dissolved in the PBS buffer. Since ammonia is highly soluble in water, we assume that the bubbles we observed in acetate buffer should be CO_2 gas because the abundant hydrogen ions in acetate buffer inhibit the dissolution of CO_2 and the ionization of carbonic acid. That’s why we only distinguished the states of CO_2 in different buffers in this conclusion. However, to make our conclusion clearer, we revised this conclusion as follows: *“The above results indicate that the urease catalysis reaction produces dissolved NH_3 and CO_2 in PBS buffer and dissolved NH_3 and CO_2 gas in acetate buffer.”*

Reviewer: “2. Page 3, lines 88: UrNMs are dispersed in phosphate buffer saline (PBS) buffer as shown in Fig.1. However, in video S3, more bubbles are generated in the acetate buffer compared to PBS buffer. This means UrNMs would have swarm behavior in the acetate buffer.”

Author’s response 2: We thank the reviewer for this valuable comment. Although ammonia dissolves in both PBS and acetate buffer, CO_2 may exist in acetate buffer (pH=4.6) because the abundant hydrogen ions inhibit the dissolution of CO_2 and the ionization of carbonic acid. Thus, in video S4, more bubbles are generated in the acetate buffer than in the PBS buffer. We state that *“The upward movement of a nanomotor swarm is due to buoyancy arising from the density difference between the reaction product-rich particulate and the media with fuel.”* Since more bubbles are generated in the acetate buffer, UrNMs are indeed supposed to have swarming behavior in the acetate buffer.

We conducted five repeated experiments to investigate the motion behavior of UrNMs in urea dissolved in acetate buffer. We prepared UrNMs and dispersed them in an acetate buffer. Subsequently, a drop of UrNMs solution (5 mg/mL) was introduced at the middle of the bottom of a chamber filled with 200 mM urea solution (in acetate buffer). The swarming behavior of UrNMs in urea in acetate buffer was recorded in video S1. A time-lapse sequence of images is depicted in Fig. R1a. The UrNMs exhibit similar collective behavior as observed

in Fig. 2a. In addition, upon quantifying the upward velocity of the UrNMs particulate, the results indicate a faster upward movement, from (1.01 ± 0.04) mm/s in PBS buffer to (1.14 ± 0.04) mm/s in acetate buffer, as shown in Fig. R1b. These findings validate our assumption that the chemical products result in a higher density difference between the particulate and the media with fuel, leading to an accelerated movement.

Figure R1. Collective movement of UrNMs in 200 mM urea that is dispersed in acetate buffer. (a) A time-lapse sequence of images that show the collective movement of UrNMs in 200 mM urea in acetate buffer. (b) Velocity analysis of UrNMs particulate in PBS buffer and in acetate buffer. The significant difference is analyzed by student's t-test: $** = P < 0.01$. $N = 5$.

Reviewer: “3. Enzyme-powered MNMs utilize enzymatic catalysis for directional mobility. Studying particle and fuel concentration uncovers insights into collective movement. In fuel-rich environments, nanomotors move against gravity. However, exploring lateral flow and other directions is vital for comprehensive understanding and broader applications.”

Author's response 3: We thank the reviewer for this valuable suggestion. We recorded videos from different moving stages of the UrNMs swarm. Trajectory tracking of UrNMs in a $22 \times 8 \times 1.6$ mm (length \times height \times width) chamber shows the spreading, sinking, and swirling stages of swarming behaviour, Figure R2b-g. These trajectories on the left and right are not perfectly symmetric due to experimental limitations. Notably, after the sinking stage, UrNMs can be rolled up by the fluid flow, as shown in the swirling trajectories of UrNMs.

[figure redacted]

Figure R2. Swarming behavior of enzymatic nanomotors viewed from the side. (a) Schematics illustrating the preparation of enzymatic nanomotors and the mechanism of solutal buoyancy resulting in swarming behavior. (b)-(g) Trajectory tracking of the UrNMs in collective movement. (b) and (f) depict one-second-long trajectories during a spreading stage. (c) and (g) show one-second-

long trajectories during a sinking stage. (d) and (e) display four-second-long trajectories during a swirling stage in urea. The blue and green color-coded trajectories indicate counterclockwise and clockwise directions of UrNMs on the left and right sides of the chamber, respectively. N=15. (h) A time-lapse sequence of images that show the directional and collective movement of enzymatic nanomotors in fuel. The fluid flow is analyzed by adding tracer particles and is shown in black arrows. Scale bar: 1 mm. (i) A time-lapse sequence of snapshots of computational results according to the assumed mechanism. The color bar indicates the nanomotor concentration, and the white arrows display the fluid velocity.

Reviewer: “4. In Figure 2c, both UrNM and MSNP velocities are plotted against UrNM concentration. Were UrNM and MSNP considered separately? Specifically, does UrNM refer to a composite of Ureas and mesoporous silica? ii. Despite MPNPs exhibiting high velocity at low concentration, how can this phenomenon be explained?”

Author’s response 4: We appreciate the reviewer for pointing out this oversight in Figure 2c. We intended to study the impact of particle concentration on velocities, meaning that we consider UrNMs and MSNPs separately. UrNM concentration refers to the concentration of urease-nanomotors. UrNMs and MSNPs should be plotted against UrNM concentration and MSNP concentration, respectively. To make it clear, we replace the “*UrNM Concentration (mg/mL)*” in Figure 2c with “*Particle Concentration (mg/mL)*”. ii. We state that “*The upward movement of a nanomotor swarm is due to buoyancy arising from the density difference between the reaction product-rich particulate and the media with fuel.*” The high upward velocity of MSNPs at relatively low particle concentration, 5 mg/mL, is due to the density difference between a particulate of MSNPs and the high concentration of fuel. The upward movement of MSNPs doesn’t exist when we increase the particle concentration to 10 mg/mL while in the same condition active enzymatic nanomotors remain relatively high upward speed. This can also be explained by the simulation results, wherein at low particle concentrations, particles with small density ($\rho=1$) exhibit upward movement driven by buoyancy.

Reviewer: “5. The velocity of UrNM and MSNP against UrNM concentration is confusing, as it implies that MSNP does not contain any UrNM. Is it suggesting that both UrNM and MSNP originate from the same complex depicted in Figure 2.c?”

Author’s response 5: Thanks for pointing it out. Figure 2c should be the velocity of UrNMs and MSNPs against UrNM concentration and MSNP concentration, respectively. It’s true that MSNPs do not contain any UrNMs. We have changed the “*UrNM Concentration (mg/mL)*” in Figure 2c to “*Particle Concentration (mg/mL)*”.

Reviewer: “In addition, the authors need to consider the following minor issues:

1. Page 1, lines 35: There is an extra space between high and efficiency.”

Author's response: We thank the reviewer for carefully checking these minor issues. The line 35 in page 1 has been revised as *“ranging from enhanced diffusion in bio-fluids and targeted delivery to cancer therapy.”*

Reviewer: “2. Page 2, lines 62: nanomotors’”

Author's response: The phrase “The enzymatic nanomotors swarming behavior” in Page 2, line 62 has been removed due to a revision of the introduction.

Reviewer: “3. UrNM and UrNMs: Are these different or are they just singular and plural, it's confusing in some cases. Please clarify.”

Author's response: The difference between “UrNM” and “UrNMs” lies solely in their singular or plural form. Generally, “UrNMs” is utilized when describing many urease nanomotors within a swarm. However, when referring to “UrNM concentration”, the singular form is preferred. To ensure clarity, we carefully reviewed the usage of both “UrNM” and “UrNMs” throughout the manuscript.

Reviewer: “4. Page 6, lines 182: Here describe the contents of video S3. However, in the video, it is observed that UrNMs and Urease are added to the cuvettes at different moments. While turbidity is observed in the first two containers. These phenomena are not described in the article.”

Author's response: We appreciate the reviewer for these suggestions. In the experiments, UrNMs and urease were added to the cuvettes at different moments but for the same durations. However, to make video S4 (video S3 in the initial manuscript) show the same time durations, we edited the video to align the starting point. The turbidity observed in the first two cuvettes is due to the larger size of UrNMs (around 400-500 nm) than urease molecules. We add descriptions for this video as follows:

“We filled cuvettes with 300 mM urea solutions that were dissolved either in PBS buffer or in acetate buffer. Then UrNMs or urease solutions were added in the cuvettes, respectively. Video S4 shows clearly the convective flow from the turbidity while for smaller urease molecules, the solutions remain transparent.”

Reviewer: “5. Page 6, lines 197: Verification of the products of UrNMs catalysis reaction accelerate(accelerates) directional movement.”

Author's response: We thank the reviewer for pointing this out. To make this caption accurate, we revise the caption to *“Verification of the UrNMs catalysis reaction products.”*

Reviewer #2:

Reviewer's report: “The authors present an experimental combined with theoretical modeling study of the dynamics of enzymatic nanomotors. The article presents intriguing findings, yet the main findings outlined in the introduction lack novelty and might depend on the specific model configuration chosen. While timely, the study's novelty is challenging to assess given recent similar studies in the literature. For instance, characterizing enzymatic nanomotors' upward directional motion as a "swarm" may not be entirely accurate, as swarming typically involves collective swirling or whirling motions observed in bacterial swarms or larger-scale swarms. The analysis appears to focus primarily on the interplay of gravity and buoyancy dynamics without delving into collective behavior. A deeper analysis of the dynamics and terminology usage may be necessary for a comprehensive understanding. I would encourage the authors to present a comparison of their findings with existing literature.”

Author's response: We thank the reviewer for the valuable suggestions. Recently there have been several reports on convection-induced swarming motors. For example, J. Zhang et al.¹ employed UV light to induce local density variations and thermal buoyancy effects, driving the micromotors to aggregate and form clusters. Similarly, M. Sun et al.² utilized an optical field to generate convective flow through the photothermal effect, allowing the colloidal collectives to drift in fluidic currents. These studies underscore the significant impact of convective flow on the collective movement of micro/nanoparticles. However, in these reported works, external light sources were used to initiate the convective flow and regulate swarm migration. In our findings, the density difference between the particle-rich particulate and surrounding fuel is sufficient to trigger convective flow in a closed chamber.

There are also some papers that reported the enzyme catalysis-induced buoyancy mechanism, known as “enzyme pump”^{3,4}. In those, the enzymatic reactions on a fixed surface convert reactants to products, which have different densities compared with the reagents. The local density variations in solutions generate a force on the fluid and form convective loops in a closed microchamber. In our research, we provide the first elucidation of the mechanisms underlying the emergent swarming behaviour of enzymatic nanomotors. Enzymes were attached onto the surface of nanoparticles and move dynamically with the fluid flow. We model a swarm of light particulates immersed in a denser fluid environment. The particulate swarm moves upward, creating a convective flow in a closed fuel-filled space. We also studied the control factors such as particle and fuel concentration and media viscosity on the convective dynamics. In this sense, our findings are novel and meaningful, serving as a bridge connecting emergent swarming behavior with its potential biomedical

applications. We made comparisons between these papers and our main findings in the introduction as follows:

“In synthetic MNMs, buoyancy-driven convection has been employed for directional motility and cargo delivery. One approach utilizes incident light to generate convective flow through the photothermal effect. This convective flow can drive TiO₂ micromotors to aggregate and form clusters⁴⁴, or enable magnetic colloidal collectives to drift using fluidic currents⁴⁵. Another approach involves enzymes fixed on a surface, which catalyze fuels, inducing density variations between the reactants and products of chemical reactions. For instance, urease-attached macroscale sheets exhibit clockwise or counter-clockwise rotation via a solutal buoyancy mechanism arising from enzyme pumps⁴⁶. Additionally, urease attached to a lipid membrane induces fluid flow and the directional transport of tracer particles⁴⁷. These studies underscore the significant implications of convective flow on the collective movement of micro/nanoparticles.

Here, the buoyancy-driven convective dynamics of swarming enzymatic nanomotors arising from the density difference between the product-rich particulate and the denser environment. We attach enzymes onto the surface of nanoparticles, which move dynamically with the fluid flow. We describe the emergent swarming behaviour of enzymatic nanomotor in three dimensions (3D) to unveil the underlying mechanisms. We model a swarm as light particulates immersed in a denser fluid environment. The particulate swarm moves upward, creating a convective flow in a closed fuel-filled space. Control factors for directional and collective mobility of enzymatic nanomotors, such as particle and fuel concentration and media viscosity, are investigated. In 3D space, convective flows develop complex patterns such as vortices. To further validate our findings, we vertically confine the system to two dimensions, where convective flow is constrained to a plane, inhibiting the possible flow configurations. The convective dynamics resemble bioconvection and provide insights into the mechanisms of swarming behaviour observed in enzymatic nanomotors.”

References:

- [1] Zhang, J. et al. Light-powered, fuel-free oscillation, migration, and reversible manipulation of multiple cargo types by micromotor swarms. *ACS Nano* **17**, 251–262 (2023).
- [2] Sun, M. et al. Bioinspired self-assembled colloidal collectives drifting in three dimensions underwater. *Sci. Adv.* **9**, eadj4201 (2023).
- [3] Song, J., Shklyae, O. E., Sapre, A., Balazs, A. C. & Sen, A. Self-propelling macroscale sheets powered by enzyme pumps. *Angew. Chem. Int. Ed.* **63**, e202311556 (2023).
- [4] Sapre, A. et al. Enzyme catalysis causes fluid flow, motility, and directional transport on supported lipid bilayers. *ACS Appl. Mater. Interfaces* **16**, 380–9387 (2024).

Reviewer: “Further, the following major points need to be considered before publication in any journal:

1. While the exploration of particle concentration, fuel concentration, viscosity, and vertical confinement is valuable, I propose conducting a control experiment to further elucidate the system's behavior. By confining the system to two dimensions where convection is not feasible, we can observe whether directed motion persists in the absence of vertical movement. This additional experiment would provide insight into the underlying mechanisms driving the nanomotor swarm's motion and help validate the findings under different conditions.”

Author's response 1: We thank the reviewer for this valuable suggestion. In fact, in the manuscript, we confined the system vertically to two dimensions, where the fluid flow was significantly reduced, Fig. 4c, Fig. S13-21. We observed that after seeding a droplet of UrNMs, these particles remained in the same pattern as when they entered the plane. PIV results also revealed that the collective movement was substantially reduced (2 $\mu\text{m/s}$ on average at time 0) compared with the less confined conditions (30 $\mu\text{m/s}$ on average at time 0). These results indicate that, as the reviewer hypothesized, the confinement hinders fluid convection and the resulting collective movement of enzymatic nanomotors, providing insight into the mechanism of the buoyancy-resulted swarming behavior of nanomotors. However, to make this claim clear, we added a supportive sentence for Fig.4: “*These results indicate that the vertical confinement controls the swarms by affecting fluid convective flows and provide insight into the buoyancy-driven swarming behavior of nanomotors.*” We also introduced these findings in the introduction as: “*In 3D space, convective flows develop complex patterns such as vortices. To further validate our findings, we vertically confine the system to two dimensions, where convective flow is constrained to a plane, inhibiting the possible flow configurations.*”

Reviewer: “2. It's indeed a pertinent question. If the authors classify the observed phenomenon as "swarming," the absence of vortex motion within the swarms raises questions about the accuracy of that classification. Vortex motion, characteristic of swarming behavior, typically involves collective swirling or whirling movements within the swarm. The lack of such motion suggests that the dynamics observed may not align entirely with traditional swarming behavior. Clarifying this discrepancy could lead to a better understanding of the true nature of the observed phenomenon and its underlying mechanisms.”

Author's response 2: We thank the reviewer for this considerable thinking. In our previous studies, e.g. *A. Hortelao, et al.*¹, the phenomenon of “swarming” can be clearly recognized from the displayed videos as it shows the formation of vortices and fronts that promptly

dissociate in the collective movement of enzymatic nanomotors. This work describes and explains the observed emerging swarming behavior from a side view mainly and the effect of confinement as explained in the comment above. This is the reason why we still define the observed phenomenon as “swarming”. After considering the reviewer’s suggestion, we analyzed the trajectories of UrNMs to show the characteristic vortex motion of swarming behavior. Trajectory tracking of UrNMs in a $22 \times 8 \times 1.6$ mm (length \times height \times width) chamber shows the spreading, sinking, and swirling stages of swarming behaviour, Figure R2b. These trajectories on the left and right are not perfectly symmetric due to experimental limitations. Notably, after the sinking stage, UrNMs can be rolled up by the fluid flow, as shown in the swirling trajectories of UrNMs. In addition, as shown in Fig. R3, we recorded the movement of a UrNMs particulate in 300 mM urea containing 2 mg/mL HA (to increase the solvent viscosity and to decrease the upward velocity). During an ascending stage, there are two counter-rotating vortices within a droplet, a characteristic hydrodynamic flow pattern.

Figure R2b. Trajectory tracking of the UrNMs in collective movement. (b) and (f) depict one-second-long trajectories during a spreading stage. (c) and (g) show one-second-long trajectories during a sinking stage. (d) and (e) display four-second-long trajectories during a swirling stage in urea. The blue and green color-coded trajectories indicate counterclockwise and clockwise directions of UrNMs on the left and right sides of the chamber, respectively. $N=15$.

Figure R3. Images of an ascending droplet of UrNMs showing the formation of two counter-rotating vortices, as indicated schematically by the white arrows. The scale bar corresponds to 0.5 mm.

Reference:

[1] Hortelao, A. C. et al. Swarming behavior and in vivo monitoring of enzymatic nanomotors within the bladder. *Sci. Robot.* **6**, eabd2823 (2021).

Reviewer: “3. The distinction between active and passive swarms, as mentioned by the author in the control experiments, lacks clarity. While both types exhibit directional motion, it remains unclear why one is labeled as active and the other as passive.”

Author’s response 3: The authors thank the reviewer for pointing this out. The upward directional movement is a result of the density difference between the particulate and the solvent with fuel (urea). Therefore, if the concentration of passive particles (particles without enzyme attached, MSNPs) is rather low, in a high concentration of fuel, the density difference could be sufficient to drive passive particulate upward. This is why in Fig. 2c, one can observe the upward movement of passive particulate in the condition of low particle concentration (5 mg/mL). However, compared with the active particles (particles with urease attached, UrNMs), passive MSNPs show slower upward velocity, Fig. 2e, g. Additionally, during the spreading stage, active nanomotors form a thin boundary layer of particle-rich fluid that continues to spread until it meets the side boundary. In contrast, passive particles form a less stable boundary layer, leading to the formation of larger falling plumes earlier, causing them to sink before reaching the side boundary (see videos S2 and S3). We assume this occurs because the products of chemical reactions make the active particles less dense, and the faster upward movement of active nanomotors creates a more dynamic environment with increased fluid flow, making it less likely to form large falling plumes.

To make it clear, we revised the manuscript to refer to “active swarms” as urease-attached nanomotors and “passive particulates” as particles without the enzyme attached. Additionally, although in certain cases, both active and passive particulates show upward movement, the active ones have more impact in real application scenarios by virtue of the enhanced velocity and catalytic capability. We clarified these advantages in the revised introduction section as follows:

“Recently, Hortelao et al.³⁶ reported the emergent swarming behaviour of enzymatic nanomotors. The urease-powered nanomotors show collective migration in urea, demonstrating the ability to swim across complex paths compared to the inactive nanomotors. Furthermore, the active collective dynamics, combined with advanced imaging technologies, position them as promising tools in the field of biomedicine. For example, swarms of radio-labelled nanobots have shown an eightfold increase in tumour penetration

and approximately a 90% reduction in tumour size during radionuclide therapy³⁷. Swarms of catalase-powered nanobots overcome and disrupt mucus layer, resulting in a 60-fold increase in mucus barrier penetration, through in vitro and ex vivo validation³⁸. Hyaluronidase and urease nanomotor swarms work synergistically for enhanced diffusion in viscous media, such as synovial fluid, paving the way for treating joint injuries³⁹. Similarly, collagenase-powered MNMs^{40,41} and urease-powered iron oxide nanomotor swarms⁴² were exploited to disrupt collagen fibers, serving as a model of the extracellular medium. This disruption facilitates cell spheroids penetration and enhances the delivery efficiency of a second swarm of nanomotors by 10-fold.”

Reviewer #4:

Reviewer's report: "This manuscript provides an investigation into the swarming behavior of enzymatic nanomotors based on urease (UrNMs) driven by buoyancy-induced convection. UrNMs exhibit upward movement in fuel mediums due to density differences, which are influenced by particle concentration, fuel concentration, and viscosity. Settlement behaviors and vertical confinement experiments further validate these phenomena. Computational modeling aligns with the experimental findings, demonstrating how adjusting parameter scan control swarm behavior. These insights contribute to the understanding of collective motion. In summary, the manuscript provides an innovative foundation for understanding the swarming behavior and underlying mechanisms of urease powered nanomotors. The result is logical and reasonable. However, to ensure compliance with the publication standards of Nature Communications, the manuscript is suggested to make some modifications as follows:

1. The manuscript mentions that "Our findings are crucial for the potential biomedical applications of enzymatic nanomotor swarms, ranging from enhanced diffusion in bio-fluids and targeted delivery to high- efficiency cancer therapy". What are the potential applications of urease-based nanomotors, and what is the current status of their development? Please provide a concise explanation about their applications and technological advancements."

Author's response 1: The self-propulsion property of enzymatic nanomotors offer advantages in movement in complex media^{1,2}, overcoming biological barrier³, on demand drug delivery^{4,5} and diagnostics in vivo^{6,7}. For example, urease-nanomotors could be used as ideal drug delivery carriers for targeting cancer cells and deliver payload on demand in response to predefined stimuli. H. Choi et al.⁷ reported that the urease nanomotors facilitate penetration to the mucosa layer of the bladder wall and prolonged retention in the bladder even after repeated urination.

Furthermore, the active collective dynamics of these nanomotors, combined with advanced imaging techniques, position them as promising tools in the field of biomedicine. We have incorporated the reported biomedical applications of enzymatic nanomotor swarms and their technological advancements into the introduction as follows:

"Micro/nanomotors (MNMs) are synthetic active devices achieving self-propulsion through converting various types of energy into mechanical motion^{23,24}. Earlier works on enzyme-powered MNMs have demonstrated the motion of single particles²⁵⁻²⁷ and small clusters²⁸, as well as proof-of-concept studies in drug delivery²⁹⁻³³ and sensing^{34,35}. Nonetheless, recent reports have shifted focus to the collective motion of these particles. Recently, Hortelao et al.³⁶ reported the emergent swarming behaviour of enzymatic

nanomotors. The urease-powered nanomotors show collective migration in urea, demonstrating the ability to swim across complex paths compared to the inactive nanomotors. Furthermore, the active collective dynamics, combined with advanced imaging technologies, position them as promising tools in the field of biomedicine. For example, swarms of radio-labelled nanobots have shown an eightfold increase in tumour penetration and approximately a 90% reduction in tumour size during radionuclide therapy³⁷. Swarms of catalase-powered nanobots overcome and disrupt mucus layer, resulting in a 60-fold increase in mucus barrier penetration, through *in vitro* and *ex vivo* validation³⁸. Hyaluronidase and urease nanomotor swarms work synergistically for enhanced diffusion in viscous media, such as synovial fluid, paving the way for treating joint injuries³⁹. Similarly, collagenase-powered MNMs^{40,41} and urease-powered iron oxide nanomotor swarms⁴² were exploited to disrupt collagen fibers, serving as a model of the extracellular medium. This disruption facilitates cell spheroids penetration and enhances the delivery efficiency of a second swarm of nanomotors by 10-fold.”

References:

- [1] Ban, W. et al. Lipase-powered asymmetric silica nanomotors with a tailored head-tail structure for enhanced mucus penetration. *Appl. Mater. Today*. **34**, 101916-101923 (2023).
- [2] Cao, Y. et al. Oral nanomotor-enabled mucus traverse and tumor penetration for targeted chemo-Sono-immunotherapy against colon cancer. *Small* **18**, 2203466-2203482 (2022).
- [3] Blanco, E., Shen, H. & Ferrari, M. Principles of nanoparticle design for overcoming biological barriers to drug delivery. *Nat. Biotechnol.* **33**, 941–951 (2015).
- [4] Mitchell, M. J. et al. Engineering precision nanoparticles for drug delivery. *Nat. Rev. Drug Discov.* **20**, 101–124 (2021).
- [5] Tezel, G. et al. Current status of micro/nanomotors in drug delivery. *J. Drug Target.* **29**, 29–45 (2020).
- [6] Ye, Z. et al. Supramolecular modular assembly of imaging-trackable enzymatic nanomotors. *Angew. Chem. Int. Ed.* **63**, e202401209 (2024).
- [7] Choi, H., Cho, S. H. & Hahn, S. K. Urease-powered polydopamine nanomotors for Intravesical therapy of bladder diseases. *ACS Nano* **14**, 6683–6692 (2020).

Reviewer: “2. Swarming behavior encompasses collective motion exhibited by self-propelled entities, often characterized by coordinated interactions and emergent patterns. In the context of enzymatic nanomotors, swarming behavior refers to the tendency of these particles to aggregate or form spatially organized groups. Understanding the underlying mechanisms driving this clustering phenomenon, such as inter-particle interactions,

environmental cues, or external stimuli, is crucial for deciphering the dynamics and functionalities of these swarms. The commonly used strategies for the assembly and control of swarms are divided into four kinds in the manuscript. One of them is "(3) combining biological microswimmers, such as sperm cells and algae, into artificial moieties as a hybrid integration." Could this phenomenon be classified as a form of swarming behavior, and if so, what specific clustering behaviors do these entities demonstrate?"

Author's response 2: We thank the reviewer for pointing out this. It has been reported that synthetic microswimmers combining biological moieties show collective behavior. For example, bio-hybrid microswimmers driven by multiple *Serratia marcescens* bacteria have a strong heading preference for moving up the L-serine gradient¹. This kind of biohybrid microswimmer shows collective chemotaxis behavior. H. Chen et al.² reported magnetic nanoparticle (MNP)-loaded probiotic *Escherichia coli* Nissle1917 (EcN@MNP). The magnetotactic and hypoxia perception of EcN@MNPs provided the microrobot collective perception and the positive migration ability to target the tumour microenvironment. The hybrid swarms show advantages when moving collectively, such as collective perception. However, these reported results didn't focus on the fundamentals of swarming behavior. Cluster formation has been studied computationally in unbounded microswimmer suspensions, which is a result of hydrodynamically induced collective motion³. However, no related clustering phenomenon of biohybrid micro/nanomotors has been reported experimentally to the best of our knowledge. Therefore, we classify bio-hybrid microswimmers as synthetic swarming systems rather than strategies for the assembly and control of swarms. The introduction has been revised as follows:

"Multiple synthetic swarming systems have been developed with inspiration from nature including: (1) applying one or multiple external forces, such as magnetic fields⁵⁻⁸, light^{9,10}, ultrasound^{11,12}, electric fields^{13,14}, (2) utilizing chemicals as signals¹⁵⁻¹⁷, (3) combining biological microswimmers, such as sperm cells and algae, into artificial moieties as a hybrid integration¹⁸⁻²⁰, (4) exploiting DNA base-pair interactions^{21,22}. These well-designed swarms show many advantages compared to single-unit functionalities, like enhanced coverage and fluid mixing, intelligent multitasking, collective chemotaxis and perception, and environmental adaptation."

References:

- [1] Zhuang, J. & Sitti, M. Chemotaxis of bio-hybrid multiple bacteria-driven microswimmers. *Sci Rep* **6**, 32135-32144 (2016).
- [2] Chen, H. et al. An engineered bacteria-hybrid microrobot with the magnetothermal bioswitch for remotely collective perception and imaging-guided cancer treatment. *ACS Nano* **16**, 6118–6133 (2022).

[3] Bárdfalvy, D., Škultéty, V., Nardini, C., Morozov, A. & Stenhammar, J. Collective motion in a sheet of microswimmers. *Commun. Phys.* **7**, 93-100 (2024).

Reviewer: “3. In my mind, there are a series of articles concerning convection-induced swarming motors from, such as the group of Prof. Guan and others. It is suggested to make a comparative description about them. In this way, readers may know more its whole story.”

Author’s response 3: Recently there have been several reports on convection-induced swarming motors. For example, J. Zhang et al.¹ employed UV light to induce local density variations and thermal buoyancy effects, driving the micromotors to aggregate and form clusters. Similarly, Z. Deng et al.² utilize near-infrared (NIR) light to induce convection flows, resulting from the temperature gradient established across the NIR-exposed and unexposed areas. The migration of swarms can be controlled by external light forces. M. Sun et al.³ utilized an optical field to generate convective flow through the photothermal effect, allowing the colloidal collectives to drift in fluidic currents. These studies underscore the significant impact of convective flow on the collective movement of micro/nanoparticles. However, in these reported works, external light sources were used to initiate the convective flow and regulate swarm migration. In our findings, the density difference between the particle-rich particulate and surrounding fuel is sufficient to trigger convective flow in a closed chamber.

We made comparisons between these papers and our main findings in the introduction as follows:

“In synthetic MNMs, buoyancy-driven convection has been employed for directional motility and cargo delivery. One approach utilizes incident light to generate convective flow through the photothermal effect. This convective flow can drive TiO₂ micromotors to aggregate and form clusters^{47,48}, or enable magnetic colloidal collectives to drift using fluidic currents⁴⁹. Another approach involves enzymes fixed on a surface, which catalyze fuels, inducing density variations between the reactants and products of chemical reactions. For instance, urease-attached macroscale sheets exhibit clockwise or counter-clockwise rotation via a solutal buoyancy mechanism arising from enzyme pumps⁵⁰. Additionally, urease attached to a lipid membrane induces fluid flow and the directional transport of tracer particles⁵¹. These studies underscore the significant implications of convective flow on the collective movement of micro/nanoparticles.”

References:

[1] Zhang, J. et al. Light-powered, fuel-free oscillation, migration, and reversible manipulation of multiple cargo types by micromotor swarms. *ACS Nano* **17**, 251–262 (2023).

- [2] Deng, Z. et al. Swarming and collective migration of micromotors under near infrared light. *Appl. Mater. Today* **13**, 45–53 (2018).
- [3] Sun, M. et al. Bioinspired self-assembled colloidal collectives drifting in three dimensions underwater. *Sci. Adv.* **9**, eadj4201 (2023).

Reviewer: “4. Blood contains water, electrolytes, proteins, hormones, cells, and waste products like urea and creatinine, which has a density of 1.050 to 1.065 g/mL and a viscosity of 3.4 to 5.3 mPa·s at 37°C. On the other hand, urine consists of water, urea, creatinine, uric acid, electrolytes, and waste products. Its density ranges from 1.003 to 1.030 g/mL, and viscosity is depending on solute concentration. Could the presence of complex components in blood or urine exert an influence on the swarming behaviors of nanomotors? Exploring the interactions between nanomotor clusters and the intricate molecular compositions of biological fluids like blood and urine would provide valuable insights into the feasibility and efficacy of nanomotor-based applications in biomedical contexts. Factors such as protein binding, pH variations, and electrolyte concentrations within these biological fluids may modulate the behavior and performance of nanomotor clusters, necessitating a thorough investigation to ascertain their functional capabilities and limitations in complex physiological environments.”

Author’s response 4: The enzymatic activity of UrNMs in simulated urine and urine from mice has been examined. In these experiments, we mixed the prepared UrNMs in the simulated urine or real urine for 30 min, then separated the UrNMs for enzymatic activity examination. Figure R4 shows that the enzymatic activity of urease decreases from (5.25 ± 0.06) U/mg to (3.56 ± 0.15) U/mg in simulated urine, and to (3.57 ± 0.20) in real urine. Although there is a significant decrease in enzymatic activity in urine for 30 min, the enzymatic activity is still relatively high, verifying the potential of UrNMs for in vivo applications. This result aligns with the findings of A. Hortelao et al.¹, who observed that urease-nanomotors remained homogeneously distributed in the bladders of mice after 45 minutes. In contrast, the control groups exhibited two different phases when fresh urine started entering the bladder. It also supports the study by Simo & Serra et al.², which demonstrated substantial accumulation of radio-labelled urease nanobots in tumors following intravesical administration for 1 hour.

Swarming behavior in other body fluids such as synovial fluid has been examined by N. Ruiz-González, et al.². Z. Yang, et al.³ verified the motion capability of urease nanomotors in blood. Ultrasmall nanomotors driven by urease remained mobility with high ion tolerance and they were able to be dispersed steadily in real body fluids. T. Patiño et al.⁴ reported that the protein corona was significantly reduced by 20% when the nanomotors were active. Moreover, the motion of the urease-powered nanomotors was not hindered. These studies

provided feasible ways for maintaining enzymatic activity and verified the motion ability of enzymatic nanomotors in biological fluids.

Figure R4. Enzymatic activity of UrNMs after treatment in simulated urine or urine from mice for 30 min. (a) The real-time UV-vis light absorbance of phenol red solutions containing 200 mM urea and treated UrNMs. In the control group, the UrNMs were mixed with PBS solution. (b) The specific enzymatic activity of UrNMs after treatment. Significant difference is analyzed by students' t-test: ***= $P < 0.001$; ns = not significant ($P > 0.05$). $N=3$.

References:

- [1] Hortelao, A. C. et al. Swarming behavior and in vivo monitoring of enzymatic nanomotors within the bladder. *Sci. Robot.* **6**, eabd2823 (2021).
- [2] Simó, C. et al. Urease-powered nanobots for radionuclide bladder cancer therapy. *Nat. Nanotechnol.* **19**, 554–564 (2024).
- [3] Ruiz-González, N. et al. Swarms of enzyme-powered nanomotors enhance the diffusion of macromolecules in viscous media. *Small* **20**, 2309387–2309403 (2024).
- [4] Yang, Z. et al. Ultrasmall enzyme-powered Janus nanomotor working in blood circulation system. *ACS Nano* **17**, 6023–6035 (2023).
- [5] Patiño, T. et al. Unveiling protein corona formation around self-propelled enzyme nanomotors by nanoscopy. *Nanoscale* **16**, 2904-2912 (2024)

Reviewer: “5. The role of temperature is paramount in enzymatic reactions, with significant implications for the movement dynamics of both passive particles and nanomotors. A comparative analysis of the movement behaviors of passive particles and nanomotors under physiological temperature conditions is recommended. This analysis would shed light on how temperature influences the kinetic properties, diffusion rates, and overall mobility of these particles, elucidating potential advantages or challenges faced by nanomotors in biological environments. Furthermore, exploring the temperature-dependent responses of

nanomotors in relation to enzymatic activity and propulsion mechanisms can provide valuable insights into their performance and functionality in biomedical applications. Integrating temperature considerations into the study of nanomotor behavior contributes to a comprehensive understanding of their behavior under realistic physiological conditions, guiding the optimization of nanomotor design for enhanced efficacy and versatility in biomedical settings.”

Author’s response 5: We appreciate the reviewer’s suggestion to integrate temperature considerations into our study. Following this suggestion, we first explored the enzymatic activity in response to temperature. The absorbance of 200 mM urea solutions containing 0.025 mM Phenol Red and 2 μ L UrNMs (130 μ g/mL) was measured at a wavelength of 560 nm for 60 min. The incubation temperature was either at room temperature (25°C) or physiological temperature (37°C). The results, shown in Fig. R5a, b, indicate that the activity of UrNMs shows no significant difference between 37°C and 25°C, with a specific enzymatic activity of (4.38 \pm 0.20) U/mg at 25°C and (4.60 \pm 0.40) U/mg at 37°C.

Following the reviewer's suggestion for a comparative analysis of the movement behaviors of passive particles and nanomotors under physiological temperature conditions, we heated the urea solution (200 mM) to 37°C. We then seeded the UrNMs or MSNPs into either the heated urea solutions or those at room temperature. As shown in Fig. R5c, there is no significant difference in the upward velocity of UrNMs or MSNPs between the 37°C urea and the 25°C urea at a swarm level. Therefore, we infer that the slight temperature change between room temperature and physiological temperature has a negligible impact on the collective movement of UrNMs, as we claimed in the manuscript, “*Since the temperature change during chemical reactions is not obvious (Fig. S3), we rule out heat effect on the upward movement. In addition, the analysis of enzymatic activity and the upward velocity of UrNMs in urea at physiological temperature (37°C) shows no significant difference compared to room temperature (25°C), Fig. S4a-c.*”

Figure R5. The impact of physiological temperature on enzymatic activity and the collective movement of UrNMs. (a) The real-time UV-vis light absorbance of phenol red solutions containing 200 mM urea and UrNMs at 25°C and 37°C. N=3. (b) The specific enzymatic activity of UrNMs at 25°C and 37°C calculated according to (a). N=3. (c) Velocity analysis of active UrNMs swarms and passive MSNPs particulates in 200 mM urea at 25°C and 37 °C, respectively. N=5. Significant difference is analyzed by students' t-test: ns = not significant ($P>0.05$).

Reviewer: “6. Is the mechanism proposed in this paper universally applicable to a wide range of enzyme-driven nanomotors developed thus far, not restricted to urease-based systems? It is imperative to investigate the generalizability of this mechanism across various types of enzyme-driven nanomotors to elucidate fundamental principles governing their propulsion and collective behaviors. This inquiry not only contributes to a deeper understanding of nanomotor design and optimization strategies but also sheds light on potential commonalities or distinctive characteristics among different enzyme-driven systems, thereby advancing the field of nanotechnology.?”

Author's response 6: Yes, this mechanism for the swarming enzymatic nanomotors should be generally applicable. We synthesized catalase-powered nanomotors (CatNMs) and observed the swarming behavior (video S1) in 100 mM hydrogen peroxide. The synthesis of catalase-attached nanomotors has been added in the Methods. Table S1 shows the amount of enzymes attached to the surface of MSNPs by Bicinchoninic acid assay. Notably, the instant chemical reaction of CatNMs in H_2O_2 (100 mM) results in a burst of O_2 bubbles, which drives the CatNMs to move upward against gravity within one second. We added the universality of the proposed mechanism in the manuscript as: “*To verify the universality of our mechanisms, we synthesized catalase-powered nanomotors (CatNMs) and observed the convective dynamics of these enzymatic nanomotors in hydrogen peroxide (H_2O_2) (Fig. S5 and video S1). Notably, the instant chemical reaction of CatNMs in H_2O_2 results in a burst of oxygen bubbles, which drives the CatNMs to move upward against gravity within one second.*”

Figure R6. Convective dynamics of CatNMs. A time-lapse sequence of images that show the directional and collective movement of CatNMs in 100 mM H_2O_2 . Scale bar: 4 mm.

Table S1. The amount of different enzymes attached to the surface of MSNPs by BCA.

Sample number	Initial enzyme added (µg/mL)	Supernatant 1 (µg/mL)	Supernatant 2 (µg/mL)	enzyme attached (µg/mL)
Urease 1	255.8	151.8	0	104
Urease 2	263.8	151.8	0	112
Urease 3	282.8	156.8	0	126
Catalase 1	380.0	172.7	0	207.3
Catalase 2	376.4	187.3	0	189.1
Catalase 3	430.9	205.4	0	225.5

Reviewer: “7. In practical scenarios such as the blood circulatory system or bladder, nanomotors frequently encounter fluidic or irregularly shaped operating environments. Therefore, it is essential to simulate and analyze the clustering behavior of these motors within fluidic or flexible materials. Such simulations not only aid in understanding the dynamic interactions and spatial organization of nanomotor clusters but also inform the design and optimization of these nanoscale systems for enhanced performance and functionality in complex biological environments.”

Author’s response 7: We thank the reviewer for this suggestion. Previously our group has reported the effect of complex paths on the collective motion of urease-powered nanomotors¹. The authors designed four polydimethylsiloxane phantoms that comprise different path shapes, straight, rectangular, curved, and a curved path with longer straight trenches. These phantoms were filled with either water or 300 mM urea solution. It turned out that these enzymatic nanomotors moved faster when fuel was present in the phantom compared to their movement in water and the differences increased when introducing complex paths. These results prove that the enhanced mobility of enzymatic nanomotors in urea becomes more relevant when the complexity of the path increases.

We believe that although the analysis of the clustering behaviour of nanomotors within fluidic or flexible materials is important, it goes beyond the scope of this paper. Our existing code in the paper is based on the finite-difference approach and works only in simple geometries such as squares or rectangles. However, implementing simulations in complex geometries is worth trying in the follow-up studies.

Reference:

- [1] Hortelao, A. C. et al. Swarming behavior and in vivo monitoring of enzymatic nanomotors within the bladder. *Sci. Robot.* **6**, eabd2823 (2021).

Point-to-point Responses to Comments

Reviewer #1:

Reviewer's major comments: Thanks to the authors for the revisions made to the introduction. After thorough review, I am not convinced using the swarm terminology in the study. Our primary concern lies in interpreting and characterizing the observed phenomena as "swarming." Swarming is typically characterized by several key features, including dynamic interactions among agents, self-organization, and emergent properties that result from local interactions rather than global control. In biological systems, swarming behavior often involves complex coordination and communication between individual agents, leading to collective decision-making, adaptive responses to environmental changes, and sophisticated task execution. In this study, the described behavior of enzymatic nanomotors appears to be driven primarily by buoyancy-induced convection rather than dynamic interactions among individual nanomotors. While the observed collective movement and convective flows are indeed intriguing, they seem to result from the physical properties of the system rather than emergent behaviors arising from interactions among the nanomotors themselves. The distinction between convection-driven aggregation and true swarming behavior is crucial, as the latter implies a higher level of complexity and interaction.

Author's response: We thank the reviewer for the suggestion. We agree that the collective motion of nanomotors in our work is driven by buoyancy, and this main characteristic does not lead to collective decision-making or sophisticated task execution. We chose the term "swarm" to describe the observed phenomenon based on the state-of-the-art reports in the field of microrobotics and micro/nanomotors¹⁻⁴, as well as the definition of swarm. The *Britannica dictionary* describes swarm as: 1) to move in a large group; 2) to be surrounded by a large group of insects, people, etc. moving together; or 3) to surround with a large group. The *Collins dictionary* describes swarm intelligence as the collective behaviour of a group of animals, especially social insects such as ants, bees, and termites, that are each following very basic rules. Based on those definitions, the entities in a swarm may not require "complex coordination". This is indeed the case in our system, where although the chemical reactions taking place on the particle surface may induce communication among individual particles⁵⁻⁸, these interactions are insufficient to change the convective dynamics driven by fluid flow, which is the major component of the observed phenomena.

Said that, since this clarification does not change the conclusions or the main claims of our paper and is more of a semantic nature, we agree with replacing "swarming behaviour" with "collective behaviour" in the appropriate locations in the manuscript and revising the title to "Collective Buoyancy-driven Dynamics of Enzymatic Nanomotors".

References:

- [1] Deng, Z. et al. *Appl. Mater. Today* **13**, 45-53 (2018).
- [2] Li, Z. et al. *Chem. Eng. J.* **468**, 143393-143401 (2023).
- [3] Hu, Y., Liu, W. & Sun, Y. *ACS Appl. Mater. Inter.* **12**, 41495–41505 (2020).
- [4] Ji, Y. et al. *Angew. Chem. Inter. Edit.* **58**, 12200–12205 (2019).
- [5] Huang, Y., Lin, Z. & He, Q. *Adv. Funct. Mater.* **34**, 2311136-2311145 (2024).
- [6] Chen, C. et al. *Angew. Chem. Inter. Edit.* **130**, 247–251 (2018).
- [7] Mou, F. et al. *Nanoscale* **8**, 4976–4983 (2016).
- [8] Huang, L., Moran, J. L. & Wang, W. *JCIS Open* **2**, 100006-1000019 (2021).

Reviewer’s comment 1: I appreciate your effort to address the mismatch between the experimental data and the manuscript’s conclusions. Your revised conclusion provides a clearer understanding by stating that urease catalysis produces both dissolved NH₃ and CO₂ in PBS buffer, and NH₃ and CO₂ gas in acetate buffer. This clarification helps reconcile the experimental results with the conclusions drawn.

Author’s response 1: We thank the reviewer for the valuable comments, which help us improve the manuscript. A detailed discussion of the mechanism in different buffers is provided in the following response.

Reviewer’s comment 2: I appreciate the clarification regarding the bubble generation differences between the acetate and PBS buffers and the subsequent investigation into the swarming behavior of UrNMs in acetate buffer. Your response and the new data provided, including the time-lapse images and velocity analysis, offer valuable insights into the effects of different buffers on the behavior of UrNMs. However, the experiments demonstrate that UrNMs exhibit collective movement in both PBS and acetate buffers, the fundamental issue remains that the observed behaviors do not strongly distinguish between different types of collective phenomena, such as true swarming versus buoyancy-driven motion. For improvement, I suggest: clarifying how the observed behaviors in different buffers can be distinguished from other types of collective phenomena; providing detailed mechanistic insights into why different buffers lead to variations in bubble generation and nanomotor behavior; and conducting additional experiments to explore other factors influencing swarming behavior and verify whether the observed phenomena in acetate buffer truly represent swarming.”

Author’s response 2: Buoyancy and the convective flow are indeed the primary factors driving the observed collective states, and the observed behaviours do not strongly distinguish between different types of collective phenomena. However, the convective dynamics of collective enzymatic nanomotors are accompanied by chemical changes,

which offer important advantages for biomedical applications. We have demonstrated that enzyme catalysis occurring on the surface of active nanomotors lead to faster upward movement of particulates (Fig. 2), an increased coverage area¹, and a change in the pH of the surrounding medium (Fig. R1). It has been reported that there are interactions and communications between those chemically powered nanomotors. However, these particle-particle interactions may not be sufficient to counterbalance fluid flow and show distinct behaviour.

We conducted new experimental studies as requested by the reviewer as follows:

1. Clarifying how the observed behaviours in different buffers can be distinguished from other types of collective phenomena. Although passive nanoparticles without enzyme can be driven by fluid flow, we have investigated the upward velocity of both active nanomotor collectives and passive nanoparticle particulates. The results indicate that the active ones show faster upward velocity and form a more stable expanding layer on the upper boundary of the chamber. To clarify the difference, we introduced a pH indicator into the solution to distinguish the motion behaviour between active nanomotors and passive MSNPs. We added phenol red to the urea solutions and recorded the video using a colour camera (Thorlabs, CS165CU). The chemical reactions that occur during the collective movement of active nanomotors lead to a pH change in the surrounding solution, resulting in a colour shift from light yellow to pink. This colour change indicates the location of the active UrNMs. For comparison, passive nanoparticles were tested in the same condition (10 mg/mL, 200 mM urea). They expand along the bottom plane at time 0, and the pH of their surrounding solution remains unchanged.

Fig. R1 Collective motion of UrNMs/MSNPs indicated by colour change. A time-lapse sequence of images shows the directional and collective movement of (a) UrNMs in urea with phenol red and (b) MSNPs in urea with phenol red. Scale bar: 4 mm.

2. Providing detailed mechanistic insights into why different buffers lead to variations in bubble generation and nanomotor behaviour. We presented the main reaction rate constants for CO₂ and NH₃ in PBS buffer and acetate buffer in table S2. In PBS buffer, ammonia dissolves in water, pH increases from 7.3 to 8.2 in 5 min, Fig. 3a. The rate constant for CO₂ dissolving in basic solutions ($k=1.21 \times 10^4 \text{ M}^{-1} \text{ s}^{-1}$) is much higher than the reverse rate constant ($k=4.0 \times 10^{-4} \text{ s}^{-1}$). In acetate buffer, the pH of the urea solution maintains at around

4.6 in 5 min, Fig. 3b. Ammonia dissolves in acidic solutions. The rate constant for CO₂ dissolving in water ($k=0.037 \text{ s}^{-1}$) is much smaller than the reverse constant of HCO₃⁻ combining with H⁺ ($k=1.24 \times 10^5 \text{ M}^{-1}\text{s}^{-1}$). Therefore, CO₂ bubbles can be observed in acetate buffer.

Table S1. The main reactions and rate constants for CO₂ and NH₃ in PBS buffer and acetate buffer.

	Main chemical reactions	Rate constant	Reverse rate constant
In PBS buffer	1. CO ₂ (aq) + OH ⁻ ⇌ HCO ₃ ⁻	^[2] 1.21 × 10 ⁴ M ⁻¹ s ⁻¹	^[2] 4.0 × 10 ⁻⁴ s ⁻¹
	2. HCO ₃ ⁻ + OH ⁻ ⇌ CO ₃ ²⁻ + H ₂ O	^[3] 6.0 × 10 ⁹ M ⁻¹ s ⁻¹	^[3] 3.0 × 10 ⁵ s ⁻¹
	3. CO ₂ (aq) + NH ₃ ⇌ NH ₂ COOH	^[4] 4.5 × 10 ² M ⁻¹ s ⁻¹	^[4] 68 s ⁻¹
In acetate buffer	4. CO ₂ + H ₂ O ⇌ HCO ₃ ⁻ + H ⁺	^[2] 0.037 s ⁻¹	^[2] 1.24 × 10 ⁵ M ⁻¹ s ⁻¹
	5. HCO ₃ ⁻ ⇌ H ⁺ + CO ₃ ²⁻	^[3] 59.4 s ⁻¹	^[3] 5.0 × 10 ¹⁰ M ⁻¹ s ⁻¹
	6. NH ₃ + H ⁺ ⇌ NH ₄ ⁺	^[5] 4.3 × 10 ¹⁰ M ⁻¹ s ⁻¹	^[5] 24.6 s ⁻¹
	7. H ⁺ + OH ⁻ ⇌ H ₂ O	^[3] 2.31 × 10 ¹⁰ M ⁻¹ s ⁻¹	^[3] 1.4 × 10 ⁻³ M s ⁻¹

3. Conducting additional experiments to explore other factors influencing swarming behaviour and verify whether the observed phenomena in acetate buffer truly represent swarming. We changed “swarming behaviour” to “collective behaviour” as we explained previously in response to the reviewer’s major comments. We clarified in the manuscript that due to the solutal buoyancy enzymatic nanomotor collectives show upward movement. Then, in the presence of an upper surface, they form a thin boundary layer of nanomotor-rich heavier fluid, which becomes unstable, leading to the formation of falling plumes. Therefore, we assume that factors impacting solutal buoyancy will influence the collective behaviour, specifically in terms of the upward velocity and the stability of upper boundary layer. We have verified that the velocity changes due to variations in particle concentration or fuel concentration. We introduced viscosity and vertical confinement to resist the fluid flow. As a result, the active particulate moved upward slowly, and the droplet began to collapse before reaching the upper boundary. The increased confinement inhibited fluid flow inside the chamber, thus affecting the dynamic collective behaviour of UrNMs.

Based on the same mechanism, other factors such as the material of nanomotors and the propulsion mechanism should have different influences on the collective behaviour. First, enzymatic nanomotors based on the chassis of metal materials, such as gold and iron, or polymer materials, such as PLGA and chitosan, will inevitably induce density changes of the nanoparticles, thus leading to different collective behaviours. Secondly, different propulsion mechanisms affect the collective behaviour. For instance, as we demonstrated in video S1 and Fig. S6, the instant chemical reaction of CatNMs in H₂O₂ results in a burst of oxygen bubbles, which drives the CatNMs to move upward against gravity within one second. Additionally, these nanomotors continuously generate bubbles, which keep them near the upper boundary and prevent the formation of large, unstable plumes. This finding is similar to what has been reported recently by Serra-Casablancas et al.⁶ the collective behaviour of catalase-nanobots.

The collective behaviour of UrNMs in urea in acetate buffer is also caused by the buoyancy and convective flow. In PBS buffer, the produced ammonia dissolves in water, reducing the density of the active particulates compared to passive ones. In acetate buffer, as previously discussed, the generated microbubbles enhance buoyancy, further reducing particulate density and promoting the collective motion of the nanomotors. Since we define the collective behaviour of enzymatic nanomotors as their collective movement driven by buoyancy and convective flow, accompanied by chemical changes in the surrounding medium, we consider the observed phenomenon in acetate buffer to be a true example of collective behaviour.

References:

- [1] Ruiz-González, N. et al. *Small* **20**, 2309387–2309403 (2024).
- [2] X. Wang, et al. *Phys. Chem. A* **114**, 1734-1740 (2010).
- [3] Y. Feng et al. *Appl. Phys. Rev.* **8**, 011406 (2021)
- [4] X. Wang et al. *J. Phys. Chem. A* **115**, 6405–6412 (2011)
- [5] M. T. Emerson, E. Grunwald, R. A. Kromhout, *J. Chem. Phys.* **33**, 547 (1960).
- [6] Serra-Casablancas, M. et al. *ACS Nano* **18**, 16701-16714 (2024).

Reviewer's comment 3: Thank you for your detailed response and for updating the figures to include additional data on the trajectories of the enzymatic nanomotors. I also appreciate the effort you have put into enhancing the visual representation of your work. However, the newly added content, specifically the images and trajectory data, does not significantly contribute to a deeper understanding of the underlying mechanisms of the observed behavior. The updated figures primarily illustrate the clockwise and counterclockwise flow patterns of the enzymatic nanomotors. While these visuals effectively demonstrate the flow directions, they do not offer new insights into the mechanisms driving these movements.

The observed phenomena appear to be a result of fluid dynamics rather than an indication of different underlying mechanisms or behaviors.

Author's response 3: We provide this clockwise and counterclockwise trajectories to describe the convective dynamics and the swirling in collective movement. Our explanation for the underlying mechanisms of the emergent swarming (or collective) behaviour is due to buoyancy arising from the density difference between the reaction product-rich particulate and the media with fuel. We state that individual nanomotors perform urease catalysis reaction and generate ammonia and carbon dioxide, making the particulate lighter and different performance between active particulates and passive ones.

Reviewer #2:

Reviewer's report: “The authors have successfully addressed all my queries, and the revised manuscript is significantly improved with respect to the scientific relevance and clarity of presentation. The manuscript is now suitable for publication.”

Author's response: Thank you very much for your valuable comments, which help us to refine our work and enhance the quality of the manuscript.

Reviewer #4:

Reviewer's report: The authors have responded to all the comments and suggestions. I am satisfied by the most of the answers. Thus, the manuscript has been greatly improved. At the same time, I have still several issues to be addressed in the following.

Reviewer's comment 1: The authors did not discuss propulsion mechanism in the revised manuscript. In fact, urease-powered motors may be propelled with different propulsion mechanisms, such as ionic diffusiophoresis and microbubble recoils depending on the urea concentration and the motor structure. Will whether the propulsion mechanism affect the swarming behavior? As the motor in this work was about 500 nm in size, similar to that reported in (ACS Nano, 2023, 17, 6023–6035), the propulsion is possible to be dominated by ionic diffusiophoresis.

Author's response 1: We thank the reviewer for the constructive suggestion. Here, we state that the primary mechanism for the self-organization of collective nanomotors in this study is buoyancy-driven convective flows, rather than the self-propulsion of individual motors. However, self-propulsion may still affect the results quantitatively, e.g., by increasing the effective diffusion coefficient of the particulate compared to the passive case.

Previous studies revealed that a molecularly unbalanced distribution of enzymes is sufficient to generate net motion^{1,2}. The UrNMs reported in this work rely on such inherent asymmetries for self-propulsion. Ionic self-diffusiophoresis is a plausible explanation as the ionic species have different diffusivities (H^+ : $9.3 \times 10^{-5} \text{ cm}^2\text{s}^{-1}$, NH_4^+ : $1.957 \times 10^{-5} \text{ cm}^2\text{s}^{-1}$, OH^- : $5.3 \times 10^{-5} \text{ cm}^2\text{s}^{-1}$, HCO_3^- : $1.19 \times 10^{-5} \text{ cm}^2\text{s}^{-1}$, and CO_3^{2-} : $0.923 \times 10^{-5} \text{ cm}^2\text{s}^{-1}$)³⁻⁵. These differences between the cations and the anions result in the generation of local electric fields and a double layer polarization around the particle, setting it in motion⁶. Therefore, increasing the ionic strength in the media may hinder the generation of a concentration gradient, thus diminish the self-propulsion of nanomotors. We assessed the motion profiles of urease-powered nanomotors in 50 mM urea with different ionic strength by adding varied amount of NaCl. Fig.R2 shows that the mean square displacement (MSD) increases linearly with time, a characteristic diffusive motion, and the diffusion coefficient of UrNMs decreased gradually with the increase in ionic strength.

Different propulsion mechanism may affect the collective behaviour. For instance, as we showed in Fig. S6, the instant chemical reaction of CatNMs in H_2O_2 results in a burst of oxygen bubbles, which drives the CatNMs to move upward against gravity within one second. Furthermore, these nanomotors keep producing bubbles, which maintain the nanomotors on the upper boundary and stop them from forming large unstable plumes. This finding is

similar to what has been recently reported by Serra-Casablancas et al.⁷ the swarming behaviour of catalase-nanobots.

Fig. R2 Motion analysis of UrNMs in urea with varying NaCl concentrations. (a) Mean square displacement (MSD) of UrNMs in 50 mM urea solutions with different NaCl concentrations (0.05, 0.5, 5, 25 and 50 mM), analyzed by tracking 20 particles. (b) The diffusion coefficient of the nanomotors at each condition was determined. Ionic self-diffusiophoresis is a plausible explanation, as the ionic species have different diffusivities. These differences between the cations and anions lead to the generation of local electric fields and double-layer polarization around the particle, initiating its motion. Consequently, increasing the ionic strength in the medium inhibits the formation of a concentration gradient, thereby reducing the self-propulsion of the nanomotors.

References

- [1] Hortelao, A. C., Carrascosa, R., Murillo-Cremaes, N., Patiño, T. & Sánchez, S., *ACS Nano* **13**, 429-439 (2019).
- [2] Patiño T., et al. *J. Am. Chem. Soc.* **140**, 7896–7903 (2018).
- [3] Arqué X., et al. *Research* **2020**, (2020).
- [4] Choi, H., Cho, S. H., & Hahn, S. K., *ACS Nano* **14**, 6683-6692 (2020).
- [5] Y. Feng et al. *Appl. Phys. Rev.* **8**, 011406 (2021).
- [6] E. Lee *Inter. Sci. Technol.* **26**, 323-358 (2019)
- [7] Serra-Casablancas, M. et al. *ACS Nano* **18**, 16701-16714 (2024).

Reviewer’s comment 2: The ureases were modified on the whole surface of the mSiO₂. In this case, how to simulate the distribution of products or fluid around these nanomotors? What will happen if the nanomotor becomes a Janus structure? Additionally, does the size of motors affect this phenomenon?

Author’s response 2: T. Patiño et al. verified the asymmetric distribution of urease on both the micro- and nano-sized SiO₂ spheres utilizing stochastically optical reconstruction

microscopy.^{1,2} This asymmetry results in a non-homogenous distribution of reaction products around the spherical motors and their self-propulsion.

As we explained in the manuscript, the underlying mechanisms of the emergent swarming behaviour (collective behaviour) is due to buoyancy arising from the density difference between the reaction product-rich particulate and the media with fuel. The Janus structure is usually achieved by the introduction of a metal shell, such as platinum, gold or iron. These metal materials will inevitably induce density changes of the nanoparticles, thus leading to different collective behaviours. Additionally, the size may also have an impact on the collective behaviour. Under the condition of the same particle density, smaller nanoparticles may be more stable than larger ones, exhibiting a longer expanding stage and fewer large plumes during the sinking stage. This is because smaller nanoparticles experience smaller gravitational force, and a larger specific surface area will result in relatively greater fluid resistance during sedimentation.

References:

- [1] T. Patiño et al. *J. Am. Chem. Soc.* **140**, 7896-7903 (2018).
- [2] T. Patiño et al. *Nanoscale*, **16**, 2904-2912 (2024).

Reviewer's comment 3: Some of the figures, such as Fig. 4, Fig. 6, and Figs. S13-S21, show very poor quality. They are difficult to read clearly. The author should optimize these images.

Author's response 3: We thank the reviewer for pointing this out. We have optimized the quality of the Figures in both the manuscript and the supporting information to ensure they are clear and easy to read. Additionally, we have adjusted the font size in Figs. 22 and 23 (previously Figs. 20 and 21) for better clarity.

Reviewer's comment 4: The biomedical applications are still briefly mentioned, but without explanation or evidence. This will limit the significance of the work.

Author's response 4: We thank the reviewer for recognizing the significance of the observed collective behaviour for biomedical applications. In the introduction section, we listed and summarized recent biomedical applications based on the swarming behaviour of enzymatic nanomotors, ranging from the bio-imaging, viscous layer penetration, targeted delivery, to tumour treatment, as follows:

*Hortelao et al.*³⁶ reported the emergent swarming behaviour of enzymatic nanomotors. The urease-powered nanomotors show collective migration in urea, demonstrating the ability to swim across complex paths compared to the inactive nanomotors. Furthermore, the active collective dynamics, combined with advanced imaging technologies, position them as promising tools in the field of biomedicine. For example, swarms of radio-labelled nanobots

have shown an eightfold increase in tumour penetration and approximately a 90% reduction in tumour size during radionuclide therapy³⁷. Swarms of catalase-powered nanobots overcome and disrupt mucus layer, resulting in a 60-fold increase in mucus barrier penetration, through in vitro and ex vivo validation³⁸. Hyaluronidase and urease nanomotor swarms work synergistically for enhanced diffusion in viscous media, such as synovial fluid, paving the way for treating joint injuries³⁹. Similarly, collagenase-powered MNMs^{40,41} and urease-powered iron oxide nanomotor swarms⁴² were exploited to disrupt collagen fibers, serving as a model of the extracellular medium. This disruption facilitates cell spheroids penetration and enhances the delivery efficiency of a second swarm of nanomotors by 10-fold.

These examples highlight the significance of the collective motion of enzymatic nanomotors in biomedical applications. Therefore, this work is crucial for understanding the collective motion of enzymatic nanomotor swarms and for exploring other possible biomedical applications in the future.

Point-to-point Responses to Comments

Reviewer #1 (Remarks to the Author):

The authors have satisfactorily addressed all my concerns, and the revised manuscript has significantly improved in terms of both scientific relevance and clarity.

Author's response: We sincerely thank the reviewer for his/her valuable comments, which help us to improve both our work and the manuscript.

Reviewer #4 (Remarks to the Author):

I fully understand that the authors have seriously replied to all the reviewer's comments, despite the lack of corresponding experimental data. Of course, if there are data supported, it will be more conducive to improving the quality of this manuscript. In addition, I would feel more gained if the author could clarify the possible specific contribution of the formation mechanism of swarming UrNMs mentioned to the future development of swarming micro/nanomotors in biomedicine applications, rather than simply enumerating the examples in biomedicine applications.

Author's response: We thank the reviewer for their suggestion to provide additional experimental data to improve the quality of this manuscript. Previous studies in our group have verified the difference between Janus structure and patchy-like structure. The motion of a single out-of-equilibrium particle arises from the asymmetric distribution of ions, which generates an ionic gradient¹. We have theoretically studied and observed that when a nanomotor becomes a Janus particle, its velocity at the single-particle level is higher compared to patchy-like motors². This can also be verified in experimental level. Janus mesoporous silica urease-nanomotors show increased diffusion coefficient compared with patchy-like mesoporous urease-nanomotors, with values of $1.10 \pm 0.06 \mu\text{m}^2/\text{s}$ ³ versus $0.74 \pm 0.13 \mu\text{m}^2/\text{s}$ (data in this manuscript) in 5 mM urea. Said that, collective swarming is still a different phenomenon and slightly different from single particle motion. If the fabrication of Janus structure implies the addition of heavier materials like Pt or Au, particles tend to sediment⁴⁻⁶, impeding the swarm to move upwards.

Additionally, concerning whether the size of the motors affects swarming behavior, our previous study demonstrated that micron-sized hollow urease motors also exhibited 3D motion⁷. However, size is not the only variable; particle mass also changes with increasing size. Based on this, we predict that different sizes will influence swarming behaviour. If particles are driven upward by buoyancy in a closed space, they will form convective flow and exhibit collective behaviour. Larger particles, being heavier and having a different specific surface area compared to smaller ones, show distinct instability during plume formation and sedimentation, though this effect is less pronounced than the impact of density. We have added a discussion in the main text to clarify the universality of the mechanism as follows:

We expect that this buoyancy-driven mechanism will also apply to other asymmetric motors, such as Janus motors. Previous studies in our group have verified the difference between the Janus structure and the patchy-like structure. The motion of a single out-of-equilibrium particle arises from the asymmetric distribution of ions, which generates an ionic gradient⁵⁴. A theoretical study has shown that Janus particles exhibit higher velocities

compared to patchy-like motors⁵⁵. Additionally, it has been reported that micron-sized hollow urease-motors present a 3D motion at a single particle level⁵⁶, thus it is expected that large populations of these particles will also show collective motion. However, if the fabrication of Janus structures involves heavier materials like platinum or gold, sedimentation may neglect buoyancy⁵⁷⁻⁵⁹, impeding the upward collective movement. These suggest that the buoyancy-driven mechanism could be universal for various types of motors across different length scales, provided that the gravitational effects would not suppress the buoyancy-driven motion.

Motion in 3D and confined spaces is important in different biomedical settings, such as the treatment of bladder cancer. We recently observed the 3D motion of urease-nanomotors in the bladder of mouse, these nanomotors form active and vigorous flocks and vortices in the presence of urea^{8,9}. However, the origin of this swarming behaviour remains unknown, making it essential to study the factors affecting 3D motion, such as motor concentration and fuel concentration. Based on the results presented in this paper, we believe that the data and conclusions of this paper will be of highly relevance for the design of future protocols in biomedical applications. Some biomedical applications of these nanomotors are currently being developed in our lab, including treatments for bladder cancer, ocular diseases, skin and joint conditions, as well as strategies for crossing biological barriers, which would benefit from these fundamental studies of swarming nanomotors. However, we agree that it is important to clarify the role of the UrNM swarm formation mechanism in future biomedical applications. We have added a discussion in the concluding paragraph, as follows:

The buoyancy-driven convective flow enables the collective movement of enzymatic nanomotors and promotes a more homogeneous particle distribution. In a fuel-rich environment, collective behavior occurs naturally due to buoyancy and chemical reactions, without requiring external forces. This buoyancy-driven dynamics can be harnessed to design future protocols for large tissue and organ volumes, such as the bladder and joints. It allows overcoming the limitations of current cancer treatments, including sedimentation and poor dispersion in small volumes, thereby facilitating mass transport, accumulation, penetration, and effective diffusivity of individual motors.

References

- [1] De Corato, M. et al. Self-propulsion of active colloids via ion release: theory and experiments. *Phys. Rev. Lett.* **124**, 108001-108006 (2020).
- [2] De Corato, M., Pagonabarraga, I., Abdelmohsen, L. K. E. A., Sánchez, S. & Arroyo, M. Spontaneous polarization and locomotion of an active particle with surface-mobile enzymes. *Phys. Rev. Fluids* **5**, 122001-122011(2020).

- [3] Ma, X. et al. Enzyme-powered hollow mesoporous Janus nanomotors. *Nano Lett.* **15**, 7043–7050 (2015).
- [4] Simmchen, J. et al. Topographical pathways guide chemical microswimmers. *Nat. Commun.* **7**, 10598-10606 (2016).
- [5] Katuri, J., Uspal, W. E., Popescu, M. N. & Sánchez, S. Inferring non-equilibrium interactions from tracer response near confined active Janus particles. *Sci. Adv.* **7**, eabd0719 (2021).
- [6] Katuri, J., Caballero, D., Voituriez, R., Samitier, J. & Sanchez, S. Directed flow of micromotors through alignment interactions with micropatterned ratchets. *ACS Nano* **12**, 7282–7291 (2018).
- [7] Arqué, X. et al. Ionic species affect the self-propulsion of urease-powered micromotors. *Research* **2020**, (2020).
- [8] Hortelao, A. C. et al. Swarming behavior and in vivo monitoring of enzymatic nanomotors within the bladder. *Sci. Robot.* **6**, eabd2823 (2021).
- [9] Simó, C. et al. Urease-powered nanobots for radionuclide bladder cancer therapy. *Nat. Nanotechnol.* **19**, 554–564 (2024).